# Estimation of Ground-Level NO₂ and its Spatiotemporal Variations in China Using GEMS Measurements and a Nested Machine Learning Model

Naveed Ahmad[1], Changqing Lin[1,*], Alexis K.H. Lau,[1,2] Jhoon Kim[3], Tianshu Zhang,[4,5] Fangqun Yu[6], Chengcai Li[7], Ying Li[8], Jimmy C.H. Fung[1,9], Xiang Qian Lao[10]

[1]Division of Environment and Sustainability, The Hong Kong University of Science and Technology, Clear Water Bay, Hong Kong, China

[2]Department of Civil and Environmental Engineering, The Hong Kong University of Science and Technology, Clear Water Bay, Hong Kong, China

[3]Department of Atmospheric Sciences, Yonsei University, Seoul, 03722, Korea

[4]Institute of Environment, Hefei comprehensive national science center, Hefei 230000, China

[5]Key Laboratory of Environment Optics and Technology, Anhui Institute of Optics and Fine Mechanics, Chinese Academy of Sciences, Hefei 230000, China

[6]Atmospheric Sciences Research Center, State University of New York at Albany, Albany, NY 12226, US

[7]Department of Atmospheric and Oceanic Sciences, School of Physics, Peking University, Beijing 100871, China

[8]Department of Ocean Science and Engineering, Southern University of Science and Technology, Shenzhen 518055, China

[9]Department of Mathematics, The Hong Kong University of Science and Technology, Clear Water Bay, Hong Kong, China

[10]Department of Biomedical Sciences, City University of Hong Kong, Hong Kong SAR, China

*Correspondence to:* Changqing Lin (cqlin@ust.hk)

**Abstract.** The major link between satellite-derived vertical column densities (VCDs) of nitrogen dioxide (NO₂) and ground-level concentrations is theoretically the NO₂ mixing height (NMH). Various meteorological parameters have been used as a proxy for NMH in existing studies. This study developed a nested XGBoost machine learning model to convert VCDs of NO₂ into ground-level NO₂ concentrations across China using Geostationary Environmental Monitoring Spectrometer (GEMS) measurements. This nested model was designed to directly incorporate NMH into the methodological framework to estimate satellite derived ground-level NO₂ concentrations. The inner machine learning model predicted the NMH from meteorological parameters, which were then input into the main XGBoost machine learning model to predict the ground-level NO₂ concentrations from its VCDs. The inclusion of NMH

significantly enhanced the accuracy of ground-level $NO_2$ concentration estimates, i.e., the R² values were improved from 0.73 to 0.93 in 10-fold cross-validation and from 0.88 to 0.99 in the fully trained model. Furthermore, NMH was identified as the second most important predictor variable, following the VCDs of $NO_2$. Subsequently, the satellite-derived ground-level $NO_2$ data were analyzed across subregions with varying geographic locations and urbanization levels. Highly populated areas typically experienced peak $NO_2$ concentrations during the early morning rush hours,

whereas areas categorized as lightly populated observed a slight increase in $NO_2$ levels one or two hours later, likely due to regional pollutant dispersion from urban sources. This study underscores the importance of incorporating NMH in estimating ground-level $NO_2$ from satellite column measurements and highlights the significant advantages of geostationary satellites in providing detailed air pollution information at an hourly resolution.

## 1 Introduction

Nitrogen dioxide ($NO_2$) is a pivotal trace gas within the atmosphere, exerting substantial influence on the ecological environment, air quality, and climate change (Myhre et al., 2013). This significance is underscored by its role as a prominent air pollutant with inhalable characteristics that pose potential health risks (Xue et al., 2023). Additionally, it serves as an essential precursor to the formation of secondary particles and ozone (Li et al., 2019). The origins of $NO_2$ are multifarious and intricate, stemming from diverse sources such as fossil-fuel-fired power plants, vehicular

emissions, industrial activities, biofuel combustion, and residential cooking (Jion et al., 2023). Natural sources encompass wildfires, soil emissions, and lightning discharges (Li et al., 2022). Concerted efforts, including the implementation of stringent emission control policies in China, have resulted in a gradual reduction of $NO_2$ concentrations (Fan et al., 2020). Despite these positive trends, severe $NO_2$ pollution issues persist due to the heavy emissions associated with China's rapid economic development, particularly in urban agglomerations (Meng et al.,

2018). The polluted regions in China continue to exhibit $NO_2$ concentrations that surpass the safety standard set by the World Health Organization (WHO) Air Quality Guidelines (AQG) (Chi et al., 2022).

While ground-based monitoring excels in accurately capturing $NO_2$ concentrations, the challenge lies in the low density and scattered distribution of observation stations (Wei et al., 2022). The inherent limitations in the geographical coverage of these stations, coupled with the elevated costs, render it challenging to effectively fulfill the requirements

for monitoring ground-level $NO_2$ concentrations across extensive regions (Kong et al., 2021). This spatial limitation introduces substantial uncertainties when endeavoring to assess the levels of exposure on a large scale (Chi et al., 2022). Satellite instruments offer continuous air quality monitoring with broad spatial coverage (Li & Managi, 2022). Satellite-retrieved vertical column densities (VCDs) of $NO_2$ have been extensively utilized to identify variations in $NO_2$ pollution and emissions of nitrogen oxides (NOx) across various regions (Cui et al., 2021; Iqbal et al., 2022; Park

et al., 2021). However, the official satellite products provide only the column amount of $NO_2$, not the ground-level concentrations (Lamsal et al., 2014). Consequently, there has been a discernible surge in scientific research focused on deriving ground-level $NO_2$ concentrations through satellite data analyses.

The $NO_2$ columns have been measured through polar sun-synchronous low-Earth-orbiting (LEO) satellite instruments (Yang et al., 2023). These LEO satellite instruments have a daily overpass time at exact locations. However, $NO_2$

pollution may vary significantly during different times of the day, driven by emissions, meteorology, and atmospheric chemistry (Shen et al., 2023). The single measurement per day from the LEO satellite instruments, typically taken around noon or in the afternoon, may lead to an underestimation of annual mean values (Qin et al., 2017). Previous studies have explored the diurnal variations of $NO_2$ by leveraging the differences in overpass times among these LEO satellite instruments (Boersma et al., 2008; Lin et al., 2010). However, these analyses are largely affected by the varied

performance of on-board monitoring sensors and unstable data pairing (Hilboll et al., 2013). This highlights the importance of using a quantitatively uniform air quality dataset with a much higher temporal resolution from a single suite of on-board monitoring sensors to provide new insights into the diurnal variation of air pollution.

The Geostationary Environment Monitoring Spectrometer (GEMS) stands as the inaugural satellite instrument launched for the explicit purpose of monitoring both gaseous and aerosol pollutants from a geostationary Earth orbit

(GEO) over Asia (Kim et al., 2020). It was launched successfully by the Republic of Korea on February 19, 2020, and entered its intended orbit on March 6, 2020. The primary objective of the GEMS mission is to provide hourly columnar measurements of critical air quality parameters, including $NO_2$, ozone, and aerosols, across the Asian region. Unlike traditional LEO satellite instruments, the GEO-based GEMS provides more frequent monitoring of the columnar concentration of air pollutants, thereby enhancing our comprehension of the diurnal variations of $NO_2$ over Asia (Yang

et al., 2023). Additionally, the data acquired through GEMS measurements show a significant improvement in spatial resolution compared to most existing LEO measurements.

Various studies have been conducted to estimate ground-level $NO_2$ concentrations from satellite measurements, leveraging their ability to cover a large spatial extent (Fan et al., 2021; Qin et al., 2020; Wu et al., 2021). The major bridge linking the VCDs of $NO_2$ with the ground-level concentration is theoretically the $NO_2$ mixing height (NMH).

Various meteorological conditions can govern the variations in the NMH (Ahmad et al., 2024). For instance, increased temperature facilitates the vertical dispersion of $NO_2$, leading to an increase in the NMH. To convert the VCDs of $NO_2$ into ground-level concentrations, studies have employed various techniques, such as air quality models, machine learning techniques, land-use regression, and geographically weighted regression (Chi et al., 2022; Lamsal et al., 2008; Wei et al., 2022; Xu et al., 2021). These conversion models have considered multiple meteorological factors, such as

temperature, humidity, and wind, along with the planetary boundary layer height (PBLH) (Chi et al., 2022; Qin et al., 2020; Wei et al., 2022).

Numerous past studies have highlighted the importance of the boundary layer structure in governing the occurrence and evolution of extreme air pollution episodes (Shi et al., 2020). A significant relationship between a surge in surface air pollutant concentrations and a shallow PBLH has been extensively reported (Miao et al., 2019; Su et al., 2020). It

has also been recognized that air pollutants aloft can play a core role in the evolution of surface extreme pollution episodes via vertical mixing (Zhang and Rao, 1999). When the top of the mixing layer reaches the aloft pollutant-rich layer during the daytime, air pollutants can be entrained downwards, which rapidly increases surface air pollutant concentrations (Zhang et al., 2016). In addition to the vertical exchange, radiative absorption and scattering by pollutants can modify the boundary layer structure and consequently affect ground-level pollutant concentrations. For

instance, high loadings of scattering pollutants can cool the air near the ground and result in a more stable boundary

layer, which further worsens air quality (Li et al., 2017). As a result, the PBLH has been used as a proxy of the NMH because of its ability to regulate near-surface pollution levels. However, as $NO_2$ may not be uniformly distributed within the planetary boundary layer, a significant difference may exist between the PBLH and NMH. It is important to develop a conversion model that directly considers the impacts of the NMH, which paves the way to refine the processes of converting satellite-derived columnar measurements into ground-level $NO_2$ concentrations (Ahmad et al., 2024).

Based on the GEMS measurements, Ahmad et al. (2024) evaluated the impacts of meteorological factors on the variations in the NMH over China and applied a machine learning method to predict the NMH from the meteorological parameters. In the present study, we developed a nested machine-learning-based model to evaluate the effects of NMH on the conversion of columnar $NO_2$ measurements to ground-level $NO_2$ concentrations. The inner machine learning model predicted the NMH from the meteorological parameters. Subsequently, the predicted NMH was incorporated into the main machine-learning model to predict the ground-level $NO_2$ concentrations from its VCDs. Furthermore, the satellite-derived ground-level $NO_2$ data were analyzed for subregions with different geographic locations and urbanization levels. This study aims to enhance our understanding of the effects of NMH on the conversion of satellite-based columnar measurements to ground-level $NO_2$ concentrations. Additionally, it seeks to enrich the information on spatial and diurnal patterns of ground-level $NO_2$ across China using the world's first geostationary environmental satellite.

## 2    Study area, data, and methodology

### 2.1    Study area

This study investigated the spatial and temporal variations in ground-level $NO_2$ concentrations using GEMS $NO_2$ VCDs and various ground measurements for 2021. The study area is illustrated in Fig. 1, covering most of China between 18°N-43°N and 103°E-123°E. Considering the varied characteristics of air pollution in different regions of China, we divided the study area into six subregions: North-western China (NWC, including Gansu, Ningxia, and Shaanxi); North China (NC, including Beijing, Tianjin, Hebei, Shanxi, and Inner Mongolia); Central China (CC, including Henan, Hubei, and Hunan); Eastern China (EC, including Shandong, Jiangsu, Anhui, Shanghai, Zhejiang, Jiangxi, Fujian, and Taiwan); South-western China (SWC, including Sichuan, Chongqing, Guizhou, and Yunnan); and South China (SC, including Guangdong, Guangxi, and Hainan). Satellite-derived ground-level $NO_2$ data were analyzed across these subregions.

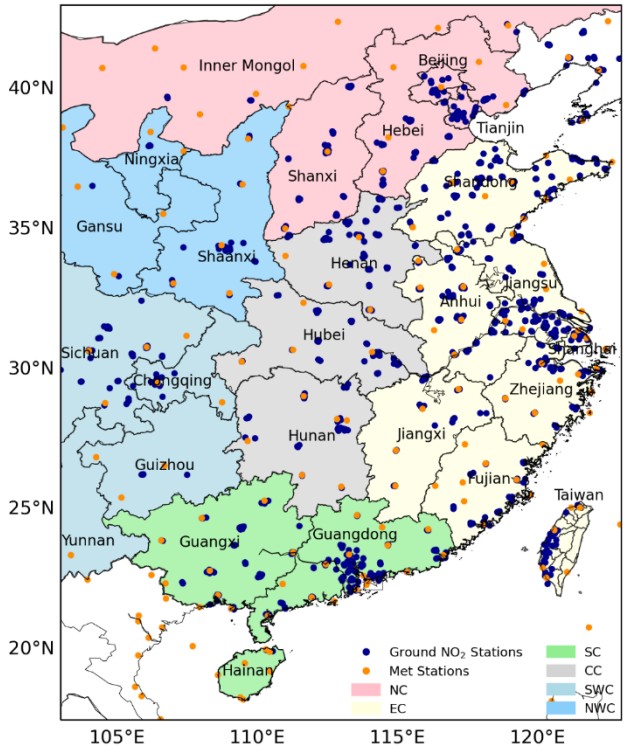

**Figure 1: Study area and six subregions shown as different background colours. Blue circles show distributions of ground-based NO₂ monitoring stations. Yellow circles show the distributions of meteorological stations.**

## 2.2 GEMS NO₂ VCDs

The GEMS NO₂ VCDs from its level 2 product were employed in this study. The NO₂ VCDs retrieval algorithm is developed based on the differential optical absorption spectroscopy (DOAS) technique (Platt et al., 2008). It initially computes slant column densities (SCDs) of NO₂ within the wavelength range of 432-450 nm. Subsequently, these SCDs are transformed into VCDs using hourly air mass factors (AMFs). The nominal detection limit for the NO₂ VCDs is $1 \times 10^{14}$ molecules/cm$^2$, with a retrieval accuracy of $1 \times 10^{15}$ molecules/cm$^2$. NO₂ VCDs surpassing the GEMS detection limit of $1 \times 10^{17}$ molecules/cm$^2$ were considered noise and consequently excluded from further analysis. The nominal spatial resolution of the GEMS NO₂ product was 7 km × 7.7 km, by binning two pixels of 3.5 km × 7.7 km each (Ahmad et., 2024). Despite the irregular shape of satellite measurement pixels due to east-to-west scans, this study performed re-gridding, which standardized the VCDs of NO₂ onto a regular grid of 0.2° × 0.4° by calculating the average of all the NO₂ VCDs within the 0.2° × 0.4° grid from 8:00 AM to 3:00 PM local time in China. Data were excluded in the presence of cloudy conditions and solar zenith angles greater than 70°. Additional information on the GEMS mission and retrieval algorithms is available in the study by Kim et al. (2020).

## 2.3 Population data

We used the latest population data for 2021 from Oak Ridge National Laboratory's (ORNL) LandScan global product (https://landscan.ornl.gov). The LandScan population data is derived through an innovative methodology that

combines geographic information science, remote sensing technology, and machine learning algorithms. Operating at a remarkably fine resolution of approximately 1 km, LandScan represents the most detailed global population distribution data accessible. As the satellite $NO_2$ measurements were on a regular grid of $0.2° \times 0.4°$, we re-gridded the LandScan population data onto a regular grid of $0.2° \times 0.4°$. The spatial distribution of population density ($D_P$, people/km$^2$) in the study area is shown in Fig. 2. Based on population density, we divided the study region into four categories: lightly populated (LP) if $D_P \leq 200$ people/km$^2$; moderately populated (MP) if $D_P > 200$ people/km$^2$ but $\leq 500$ people/km$^2$; highly populated (HP) if $D_P > 500$ people/km$^2$ but $\leq 1000$ people/km$^2$; and supremely highly populated (SHP) if $D_P > 1000$ people/km$^2$. Satellite-derived ground-level $NO_2$ data were analyzed across subregions with varying urbanization levels.

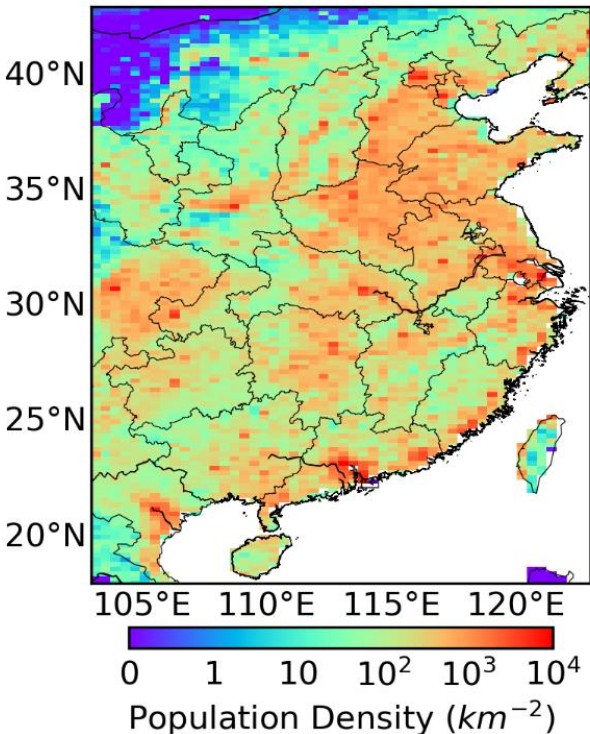

**Figure 2: Spatial distribution of population density ($D_P$, people/km$^2$) within the study area.**

### 2.4 Ground-based NO₂ and meteorological measurements

In this study, we acquired hourly $NO_2$ concentration data for 2021 from ground air quality monitoring networks situated within the study region. The spatial distribution of 856 ground-based $NO_2$ stations, sourced from the China National Environmental Monitoring Center (http://www.cnemc.cn) and the Taiwan Environmental Protection Administration (http://210.69.101.63/taqm/en/default.aspx), is shown as blue circles in Fig. 1. Meteorological variables encompassing temperature (T), air pressure (P), wind speed (WS), relative humidity (RH), dew point (DP), visibility (VIS), and precipitation (PRECIP) were used in this study. These meteorological parameters were acquired

from the global telecommunications system of the World Meteorological Organization. The spatial distribution of 208 meteorological stations is illustrated as yellow circles in Fig. 1.

**2.5 Locations matching between different datasets**

Satellite measurements, characterized by their extensive spatial coverage, stand in contrast to the localized nature of ground measurements available at specific locations. To establish a correspondence between satellite measurements and ground air quality monitoring networks, the satellite $NO_2$ data specific to the geographical coordinates corresponding to ground stations were meticulously extracted. Notably, the locations of meteorological stations may differ from those of air quality monitoring stations. Therefore, meteorological data were assigned to air quality monitoring stations situated within a 50 km radius of the meteorological station. The filtering process for model training involved the selection of stations with valid observations for all meteorological and air quality variables. These station-based datasets were used to train the machine-learning model. For predicting ground-level $NO_2$ concentrations from satellite measurements, all meteorological variables were mapped onto a regular grid of $0.2° \times 0.4°$ using the bilinear interpolation method. The spatial interpolation results of these meteorological parameters, together with the satellite measurements on the same regular grid, were employed to estimate ground-level $NO_2$ concentration at a resolution of $0.2° \times 0.4°$.

**2.6 Nested machine learning model to consider the effects of NMH**

Machine learning models have been successfully employed in estimating ground-level $NO_2$ concentrations using satellite data, typically following a two-fold procedural framework. Initializing this process involves the construction of a regression model, which is conventionally utilized to establish the overarching relationship between ground-measured $NO_2$ and its influencing factors (Chen et al., 2019; Chi et al., 2022). In this phase, the sample data undergoes division into a training dataset and a test dataset for model training and subsequent verification, respectively. The attainment of an optimal regression model is facilitated through parameter optimization techniques. Subsequently, the second phase entails the application of the regression model, where relevant data is inputted for application analysis to estimate the results.

Within machine learning studies, the ensemble learning paradigm emerges as a prevailing methodology to amalgamate diverse learning algorithms into a cohesive regression model characterized by robust performance across multifaceted domains. Owing to the disparate methodologies employed in the generation of individual learners, ensemble learning bifurcates into two principal categories: the sequential instantiation of individual learners, as encapsulated by the boosting approach, and the concurrent instantiation of individual learners, exemplified by bagging and Random Forest (Friedman et al., 2000; Prasad et al., 2006). The boosting algorithm, a variant of the lifting technique, is instrumental in diminishing variance in supervised learning scenarios, wherein distinct models are formed through the employment of disparate loss functions. XGBoost leverages both first-order and second-order derivatives to enhance the precision of model loss, a strategy that proves instrumental in achieving higher accuracy. Notably, during the process of selecting the optimal splitting point, XGBoost facilitates parallel optimization. This concurrent optimization significantly mitigates computational complexity, thereby effectively curtailing overfitting tendencies in the model. XGBoost

stands out as a notably efficient end-to-end gradient boosting tree framework, adept at transforming numerous weak learners into robust ones through boosting. This framework frequently demonstrates reduced computational overhead and enhanced predictive accuracy when compared with alternative ensemble tree models (Chen and Guestrin, 2016). Moreover, XGBoost exhibits a lower susceptibility to overfitting by mitigating the bias within the context of bias-variance decomposition. XGBoost has been empirically demonstrated to adeptly capture nonlinear relationships between predictions and predictors, yielding precise estimations through its regularized boosting methodology. This approach constructs the ultimate model by iteratively refining simpler and weaker models, each subsequent tree learning from its predecessors and updating residual errors via gradient descent to optimize the loss function. Within the XGBoost framework, an augmented penalty term is incorporated into the error function to fine-tune the objective function, thereby smoothing the final learned weights and mitigating overfitting tendencies. Additionally, to further mitigate overfitting, feature sub-sampling and shrinkage techniques are integrated (Liu 2021). The study by Van et al. (2022) also demonstrated the XGBoost algorithm as the most suitable lightweight algorithm based on the comparative analysis of three machine learning models, i.e., XGBoost, Decision Tree, and Random Forest. The XGBoost algorithm has proven to be useful in various air quality studies, including those focusing on the conversion between satellite-based column measurements and ground-level concentrations (Shao et al., 2023; Zhao et al., 2023). More details on the XGBoost regression model can be found in Chi et al. (2022).

In this study, a nested XGBoost machine learning model was developed to incorporate the NMH to convert columnar measurements into ground-level $NO_2$ concentrations. The schematic illustration of the nested XGBoost machine learning model implemented in this study is depicted in Fig. 3. Firstly, an inner machine learning model (i.e., random forest) was applied to predict the NMH using meteorological variables as input parameters. The evaluation of the predicted NMH showed a good agreement with the measurement-based results, with respective coefficient of determination ($R^2$) values of 0.84 and 0.96 for the 10-fold cross-validation and fully trained model (Ahmad et al., 2024). The NMH dataset was then mapped onto a regular grid of $0.2º \times 0.4º$ and incorporated into the main machine learning model (i.e., XGBoost regression) to estimate ground-level $NO_2$ concentrations. The main XGBoost machine learning model employed eleven input parameters, including GEMS $NO_2$ VCDs, NMH, two temporal variables (i.e., month of the year and hour of the day ranging from 8 AM to 3:00 PM), and seven meteorological parameters (i.e., T, P, WS, RH, DP, VIS, and PRECIP). The months are numbered from 1 to 12, corresponding to January through December, exactly as per the real months of the observations. All common meteorological variables available from the ground monitoring network were used in this study. The ability of these meteorological variables to regulate near-surface $NO_2$ levels is ranked by feature importance in the machine learning model. In our previous study, these meteorological parameters were shown to impact the vertical mixing of $NO_2$ to varying extents (Ahmad et al., 2024). For instance, elevated temperatures are conducive to the upward mixing of air pollutants. Increased wind speed is associated with an unstable atmosphere and can impact $NO_2$ levels by modifying the vertical dispersion and horizontal transport of air pollutants. Increased surface air pressure often leads to large-scale sinking air motion, which suppresses the vertical dispersion of $NO_2$. In this study, all input parameters were filtered based on available satellite observations for the year 2021. To reveal the impacts of the NMH, we compared the performance of the basic XGBoost machine

learning model without considering the NMH (Model I) and the nested XGBoost machine learning model after considering the NMH (Model II).

To avoid overfitting and assess the efficacy of the model, the 10-fold cross-validation methodology was employed. The dataset was partitioned into 10 groups of comparable size, with nine folds utilized for model fitting. The remaining fold served as a validation set to gauge model performance. This iterative process was repeated ten times, with each fold serving as the validation set, to evaluate the model's performance across all folds comprehensively. A set of widely recognized statistical metrics, including $R^2$, root mean squared error (RMSE), mean deviation (MD), and mean absolute percentage error (MAPE), were adopted to quantify the model's performance. In addition to the cross-validation, the XGBoost regression model was trained using the entire dataset of input parameters to predict the ground-level $NO_2$ concentrations on a regular grid of $0.2^o \times 0.4^o$ across the study region for the year 2021. The fully trained model was assessed using the same statistical indicators to evaluate its predictive performance comprehensively.

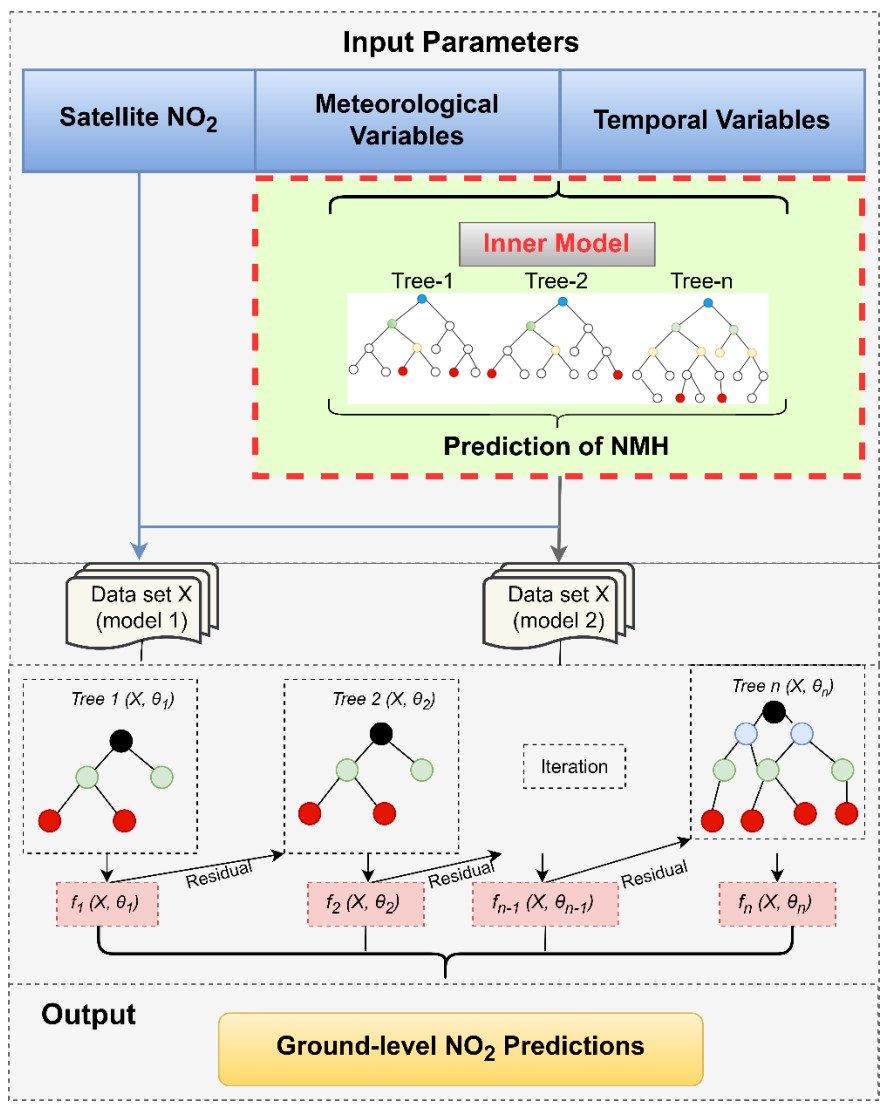

**Figure 3: Schematic diagram of the nested XGBoost machine learning model, including an inner model to predict the NMH from meteorological values and main XGBoost regression model to convert the column measurements into ground-level NO₂ concentrations. The basic XGBoost model (Model I) does not consider NMH from the inner model and utilizes only ten input variables for testing and training, namely: satellite NO₂, two temporal variables, and seven meteorological variables. The nested XGBoost model (Model II) considers the NMH from the inner model as an additional input variable, along with the other ten input variables used for the basic model. Therefore, the nested model utilizes eleven input variables for testing and training: satellite NO₂, two temporal variables, seven meteorological variables, and the NMH predictions from the inner model.**

## 2.7 Hourly, seasonal, and annual correction factors

There was some missing data for satellite $NO_2$ VCDs due to cloudy conditions between 8:00 AM and 3:00 PM for 2021. Therefore, we applied the correction factors, representing the ratio between the average $NO_2$ from all ground measurements and the average ground $NO_2$ measurements when satellite data was available (Eq. 1). These correction factors were used to obtain a bias-corrected estimation of satellite-derived ground-level $NO_2$ concentrations for each hour from 8:00 AM to 3:00 PM.

$$F(k) = \frac{\frac{1}{m}\sum_{i=1}^{m} C_g(i,k)}{\frac{1}{n}\sum_{i=1}^{n} C_g(i,k)} \tag{1}$$

Here, $F(k)$ represents the correction factor for hour k (each hour from 8:00 AM to 3:00 PM), $C_g$ represents ground-measured $NO_2$ concentrations, $m$ shows all ground measurements of $NO_2$, and $n$ corresponds to ground measurements of $NO_2$ only when the satellite data was available. For a specific hour, the maximum possible value of $m$ index in Eq. 1 is 365 for one year. The station-based spatial distributions of correction factors for each hour from 8:00 AM to 3:00 PM are shown in Fig. S1. As the predicted $NO_2$ concentrations in the study region were on a regular grid of 0.2° × 0.4°, the bilinear interpolation was applied to map the correction factors for each hour from 8:00 AM to 3:00 PM on the same regular grid of 0.2° × 0.4 ° (Fig. S2). The bias-corrected ground-level $NO_2$ concentrations for each hour from 8:00 AM to 3:00 PM were then estimated using Eq. 2.

$$C_s(k) = C_{s,0}(k) \times F(k) \tag{2}$$

where $C_s(k)$ represents the bias-corrected satellite-estimated ground-level $NO_2$ concentrations for the hour k, $C_{s,0}(k)$ represents initial predicted $NO_2$ concentrations.

Further, as the satellite data was available only during the daytime from 8:00 AM to 3:00 PM, there was also missing satellite data for nighttime and other hours of the day beyond 8:00 AM and 3:00 PM. Therefore, for seasonal correction factors, we calculated the ratio between the seasonal average of all available ground-measured $NO_2$ concentrations for 24 hours and the seasonal average of ground-measured $NO_2$ when the satellite data was available. The station-based and interpolated spatial distributions of correction factors for each season (i.e., spring, summer, fall, and winter) are presented in Fig. S3. Subsequently, Eq. 2 was used to calculate the bias-corrected ground-level $NO_2$ concentrations for each season. Similarly, to obtain the annual correction factor, we estimated the ratio between the annual average of all available ground-measured $NO_2$ concentrations for 24 hours and the annual average of ground-measured $NO_2$ when the satellite data was available (Eq. 3).

$$F = \frac{\frac{1}{j}\sum_{i=1}^{j} C_g(i)}{\frac{1}{p}\sum_{i=1}^{p} C_g(i)} \qquad (3)$$

Here, $F$ represents the annual correction factor, $C_g$ represents ground-measured $NO_2$ concentrations, $j$ shows all ground measurements of $NO_2$, and $p$ corresponds to ground measurements of $NO_2$ only when the satellite data was available. For the annual correction factor, the maximum possible value of $j$ index in Eq. 3 is 8760 for one year. The spatial distributions of station-based and interpolated annual correction factors are shown in Figs. S4. Then, Eq. 2 was

used for the bias correction of annual ground-level $NO_2$ concentrations.

**3 Results**

**3.1 Evaluations of the nested XGBoost machine learning model and its feature contribution**

The basic XGBoost model, referred to as Model I, was trained and evaluated by considering GEMS $NO_2$ VCDs together with temporal and meteorological variables as input parameters. Then, the nested XGBoost model, referred

to as Model II, was trained and evaluated by considering the NMH as input parameters in addition to the input parameters of Model I. Fig. 4a shows the 10-fold cross-validation of Model I. It depicts a value of 0.73 for $R^2$, while the RMSE, MD, and MAPE were 8.06 $\mu g/m^3$, 0.09 $\mu g/m^3$, and 39.68%, respectively. The 10-fold cross-validation of Model II after considering the NMH is revealed in Fig. 4c, which shows an improved $R^2$ value of 0.93 and a lower RMSE of 4.19 $\mu g/m^3$, MD of 0.01 $\mu g/m^3$, and MAPE of 14.78%. Further, we trained Model I and Model II on the

entire dataset of the input parameters for the year 2021. The evaluations of the fully trained Model I and Model II are presented in Fig. 4b and Fig. 4d, respectively. Again, Model II shows a lower bias and an improved $R^2$ value after considering the influences of NMH (e.g., $R^2$ increases from 0.88 to 0.99). These results clearly demonstrate that the inclusion of NMH has a great influence on the model's performance. By adding NMH as an input parameter to the machine learning model, it can better capture the vertical distributions of $NO_2$ and hence can predict the ground-level

$NO_2$ concentrations with higher accuracy and lower bias. Given the superior performance of Model II in accurately predicting ground-level $NO_2$ concentrations, we used the predictions from Model II for further analysis in this study.

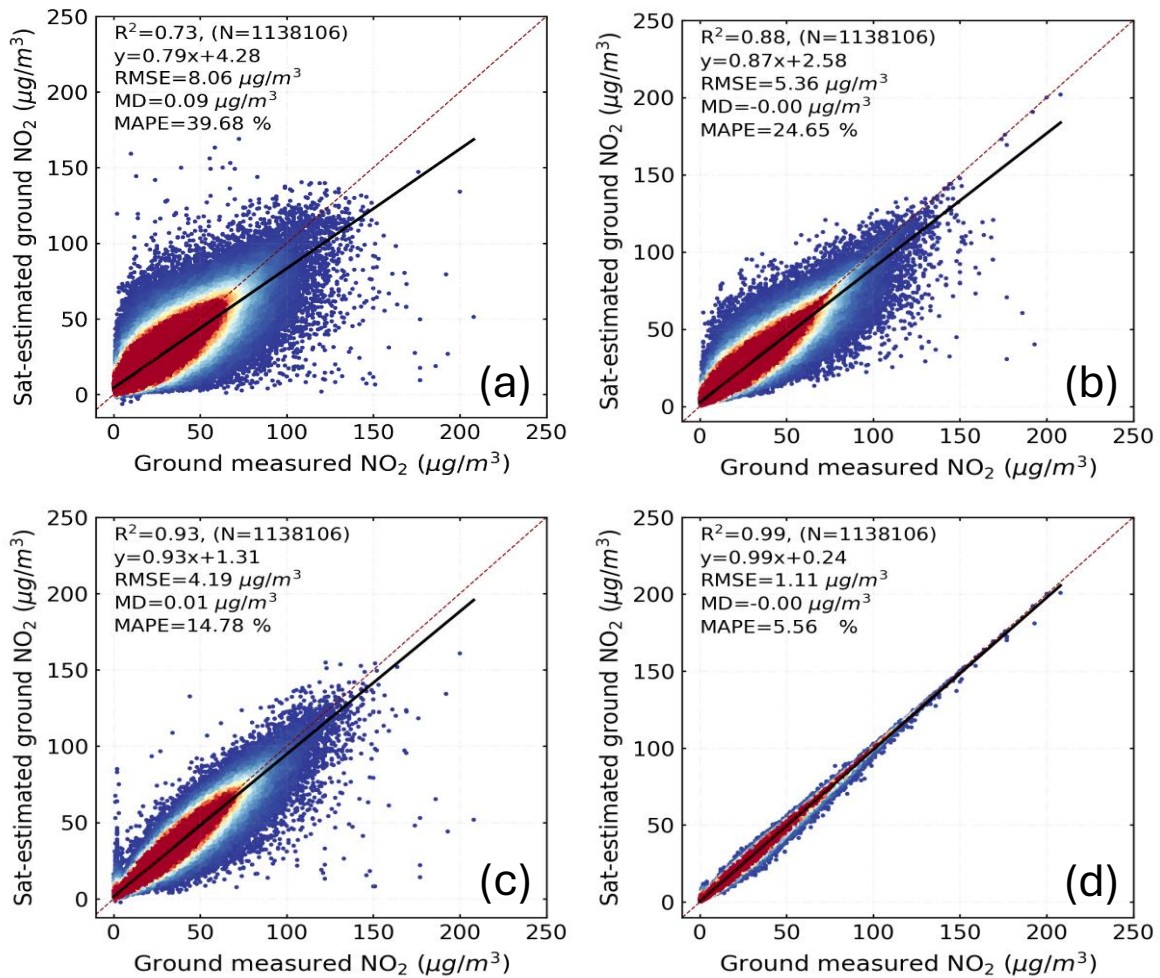

**Figure 4: The 10-fold cross-validation (a) and the validation of a fully trained model (b) for satellite-estimated ground-level NO₂ concentrations from basic Model I without considering the NMH. The 10-fold cross-validation (c) and the validation of a fully trained model (d) for satellite-estimated ground-level NO₂ concentrations from nested Model II after considering the NMH. The red dotted line represents a 1:1 relationship. The solid black line is the line of best fit between the ground-measured NO₂ and the satellite-estimated NO₂. The scattered dots represent the individual NO₂ values for each ground measurement and satellite-based estimation. The color scale ranging from red to blue represents the density of the NO₂ values, with red indicating high density and blue representing low density.**

A total of 11 features were involved in the predictions of ground-level NO₂. These features include GEMS NO₂ VCDs, NMH, two temporal variables (hour of the day and month of the year), and seven meteorological variables (T, P, WS, RH, VIS, DP, and PRECIP). Based on the XGBoost machine learning model, the feature contribution of input parameters in descending order is presented in Fig. 5. GEMS NO₂ VCDs were identified as the top predictor variable with a feature importance of 54.98 %. The second important predictor was NMH, with a contribution of 25.64 %. The temporal variables were ranked third and fourth, with an importance of 3.23 % and 3.21 % for the month of the year and hour of the day, respectively. They were followed by the meteorological parameters with a contribution of 2.45 % from temperature, 2.23 % from visibility, 2.01 % from relative humidity, 1.86 % from pressure, 1.84 % from wind

speed, 1.63 % from precipitation, and 0.92 % from dew point. Among the predictors, the dominant contributors to the predictions were GEMS $NO_2$ VCDs and NMH, accounting for 80.62% of the predictive power. Temporal variables made a modest contribution of 6.44 %, while meteorological parameters contributed only 12.94 % to the overall prediction accuracy.

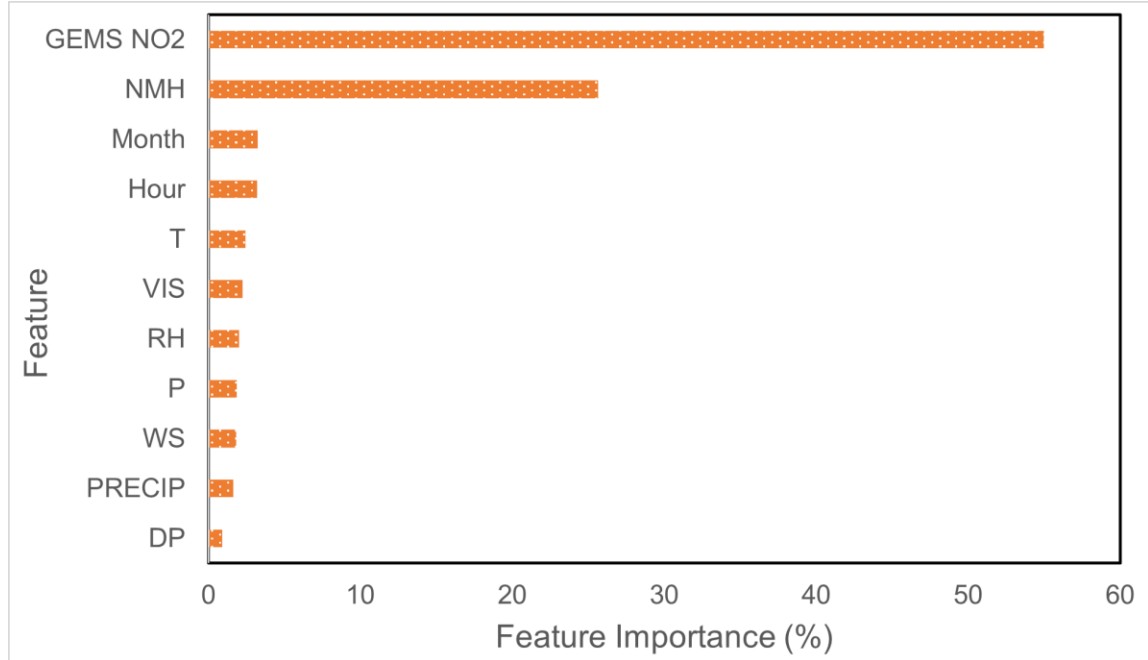

**Figure 5: Relative importance of individual input features (i.e., GEMS $NO_2$ VCDs, NMH, temporal variables, and meteorological parameters) in the XGBoost machine learning model.**

The Shapley additive explanations (SHAP) values presented in Fig. 6 were estimated from the XGBoost machine learning model to understand the impacts of individual input variables on the model's predictions. The analysis reveals that higher values of GEMS $NO_2$ VCDs correspond to higher predictions of ground-level $NO_2$ concentrations. In comparison, lower values of GEMS $NO_2$ VCDs result in lower predicted levels of ground-level $NO_2$. Conversely, lower NMH values are associated with higher predicted ground-level $NO_2$ concentrations, whereas higher NMH values are linked to lower predicted ground-level $NO_2$ concentrations. For temporal variables, the month of the year indicates the intra-annual pattern of ground-level $NO_2$, with lower concentrations observed in warm seasons and higher concentrations in cold seasons. On the other hand, the hour of the day indicates the diurnal variations of ground-level $NO_2$ values, with higher concentrations occurring during the morning and lower values during the afternoon. However, it is noted that the SHAP values for the meteorological variables, including temperature, are all small, clustered around zero, and have limited influence on the prediction results. The major and distinct impact on the model's performance for predicting ground-level $NO_2$ concentrations is observed for GEMS $NO_2$ VCDs and NMH.

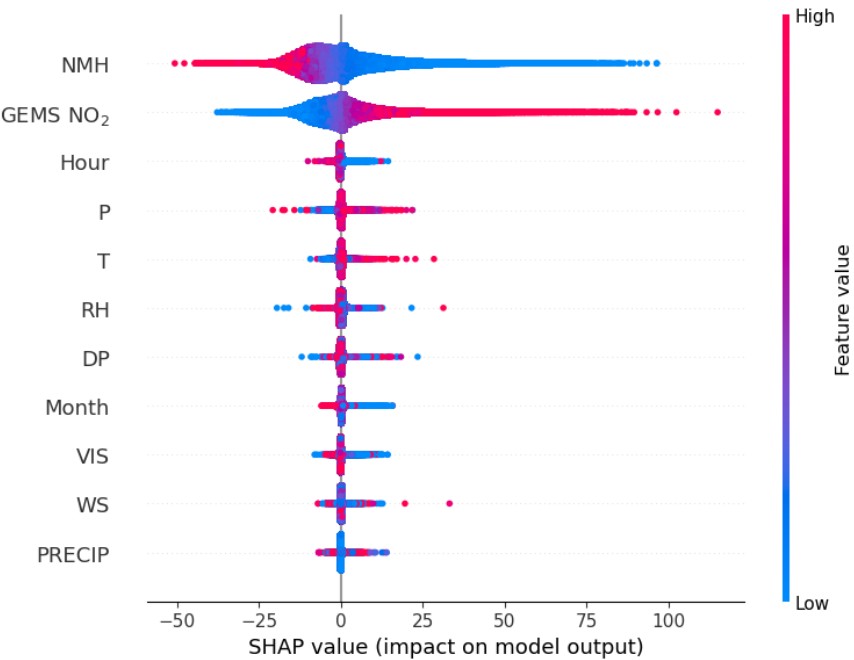

**Figure 6: Shapley additive explanations (SHAP) values from the XGBoost machine learning model to explain the impacts of individual input variables on the model's prediction of ground-level NO₂ concentrations.**

**3.2 Spatial distributions of ground-level NO₂ concentrations**

Based on the satellite-derived ground-level NO₂ concentrations (mentioned as ground-level NO₂ concentrations from hereon), Fig. 7 shows an example of the spatial distributions of ground-level NO₂ concentrations for each hour from 8:00 AM to 3:00 PM on September 29, 2021. The figure depicts a notable diurnal pattern of ground-level NO₂, with the highest values observed at 8:00 AM and lowest values observed at 3:00 PM, following a decreasing trend from

350 8:00 AM to 3:00 PM. A few GEMS NO₂ VCDs were missing due to high cloud fractions during some hours. Additionally, it should be noted that satellite measurements are only available during the daytime. We employed correction factors based on ground measurements to address the data missing issues resulting from clouds and temporal gaps (see Sec. 2.7).

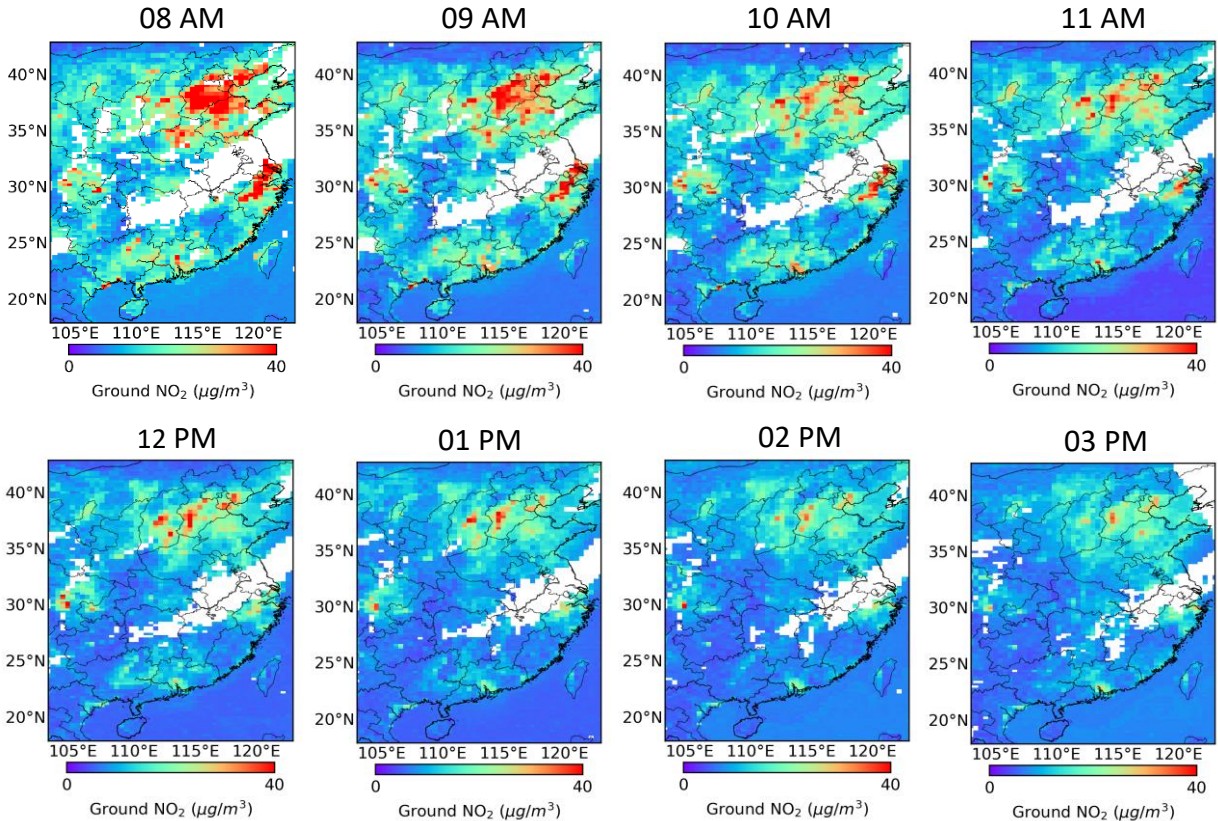

**Figure 7: Spatial distributions of the satellite-derived hourly ground-level NO₂ concentrations on September 29, 2021, for each hour from 8:00 AM to 3:00 PM.**

The bias-corrected ground-level NO₂ concentrations were applied in the further analyses. Figure 8 shows the spatial distributions of the annual average ground-level NO₂ concentrations for the year 2021 across the study region, including four urban agglomerations: Beijing-Tianjin-Hebei (BTH), Yangtze River Delta (YRD), Pearl River Delta (PRD), and Sichuan Basin (SCB). Most urban agglomerations depicted ground-level NO₂ concentrations around 40 μg/m³ or even higher. The highest ground-level NO₂ concentrations were observed in the BTH region, with a spatial distribution characterized by higher values in the region's central, southern, and southeast parts, and lower concentrations in the northern and southwestern parts. In the YRD region, elevated values were observed over Shanghai, the southern part of Jiangsu, and the northern part of Zhejiang. The PRD region exhibited the highest ground-level NO₂ concentrations in its central region, along with Guangdong's coast and central areas. In the SCB, the western part of Chongqing depicted the highest ground-level NO₂ concentrations, which can be attributed to its large population and higher emissions. The presence of a few scattered clusters of NO₂ pollution in the SCB could be attributed to economic factors and the influence of topography (Li et al., 2023). These spatial patterns are in good agreement with previous studies conducted using LEO satellite instruments (Chi et al., 2022; Qin et al., 2020; Wei et al., 2022; Wu et al., 2021; Xu et al., 2021).

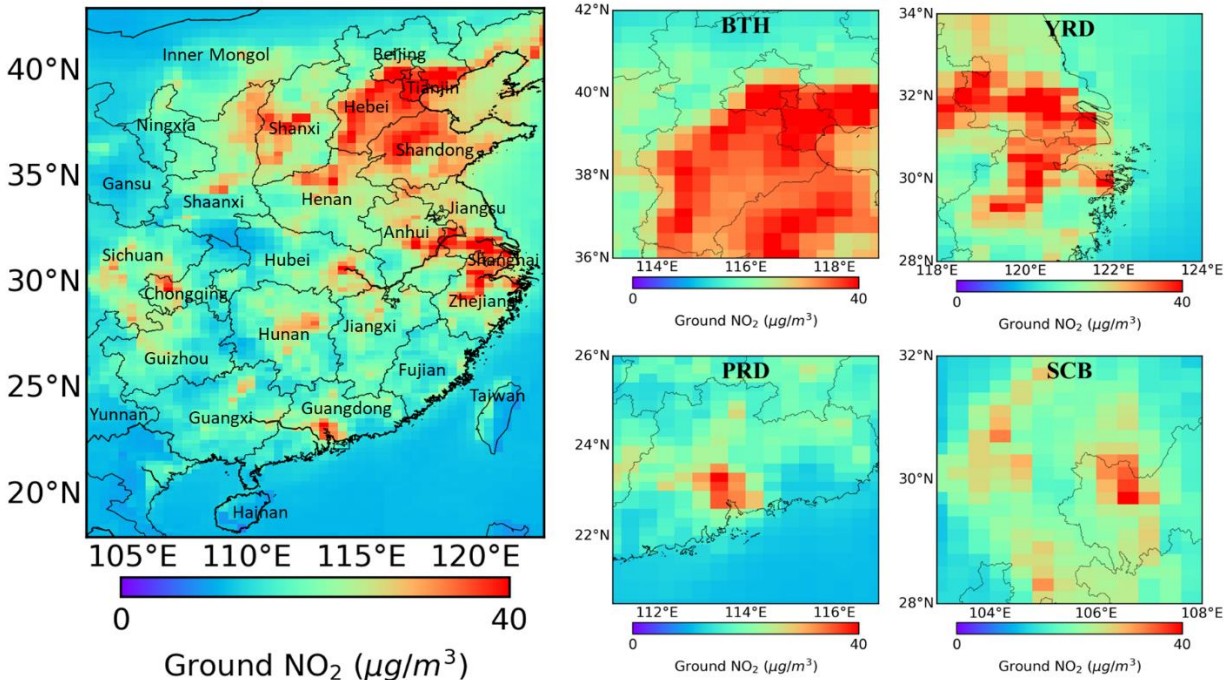

**Figure 8: Spatial distributions of annual average ground-level NO₂ concentrations for 2021 derived from satellite measurements in the study region (left panel) and in the four major urban agglomerations in China (right panel): Beijing-Tianjin-Hebei (BTH), Yangtze River Delta (YRD), Pearl River Delta (PRD), and Sichuan Basin (SCB). This annual average concentration represents the 24-hour average throughout the year of 2021 after the bias correction for the missing data issue.**

Considering the human health risks associated with NO₂, we evaluated the population exposure levels for different provinces in the study region. The provincial-level NO₂ concentrations were estimated from the annual average ground-level NO₂ concentrations. Figure 9 compares the spatial mean and population-weighted mean of NO₂ concentrations for individual provinces in descending order by the population-weighted mean. The population-weighted mean NO₂ concentrations were consistently higher than the spatial mean NO₂ concentrations, indicating that relying solely on the spatial mean may underestimate the population exposure level. The underestimation of population exposure levels using the spatial mean was more pronounced in provinces with centralized populations (e.g., Hebei and Guangdong).

The population in Tianjin province was exposed to the highest levels of NO₂, with a population-weighted NO₂ mean of 40.26 μg/m³. This level of exposure is close to the WHO Interim Target 1 (IT-1) of 40 μg/m³. The NO₂ exposure level of people living in Hebei, Shanghai, Shandong, and Jiangsu exceeded the IT-2 levels of 30 μg/m³. The NO₂ exposure levels for Beijing and Zhejiang were slightly under the IT-2 levels, with population-weighted means of 28.86 μg/m³ and 28.25 μg/m³, respectively. Residents in Henan, Anhui, Shanxi, Hubei, Sichuan, Hunan, and Jiangxi provinces were exposed to NO₂ levels exceeding the IT-3 levels of 20 μg/m³. All provinces depicted population exposure levels of NO₂ exceeding the Air Quality Guidelines (AQG) levels of 10 μg/m³. Hainan Province had the

lowest population-weighted mean $NO_2$ concentrations of 10.57 μg/m³, which closely approached the levels set by the AQG.

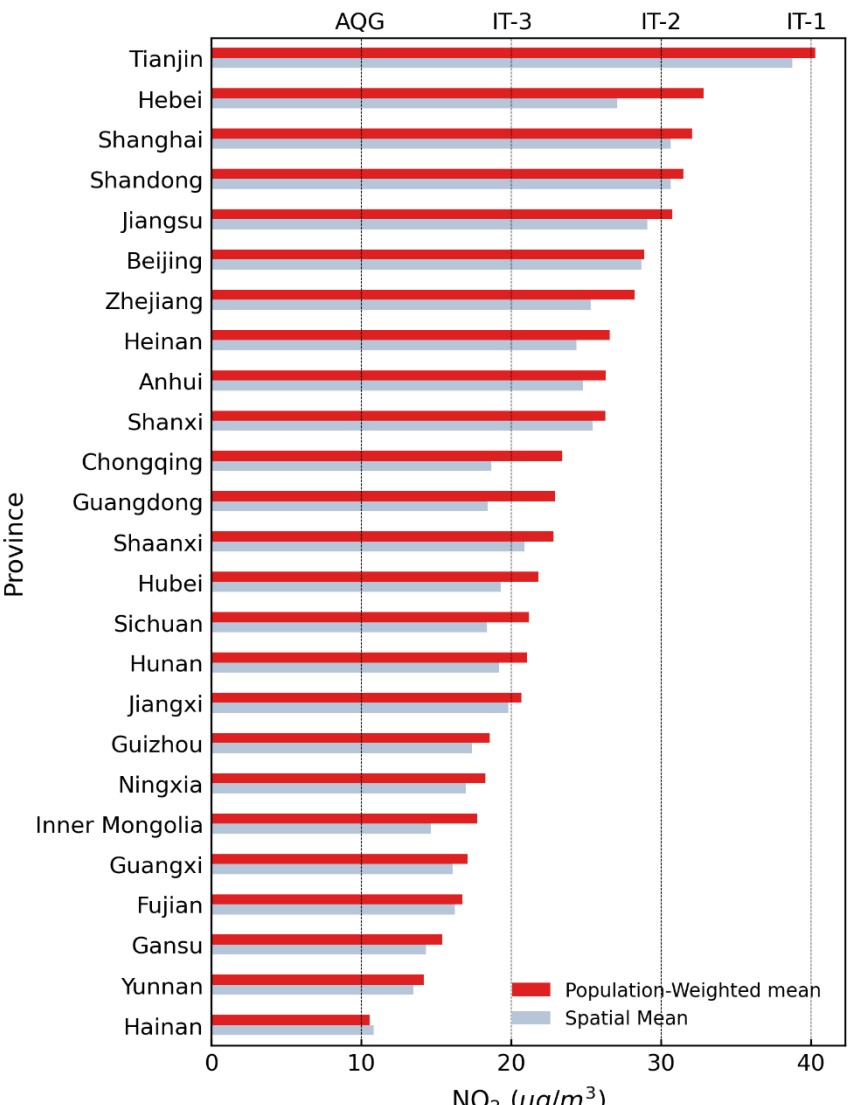

**Figure 9: Spatial mean and population-weighted mean ground-level NO₂ concentrations for 2021 in different provinces of China in the study region.**

The annual average ground-level $NO_2$ concentrations were further evaluated for all subregions with different geolocations and urbanization levels. Results are presented in Fig. 10. Overall, the highest $NO_2$ concentrations were observed in NC, followed by EC, CC, NWC, SWC, and SC. Additionally, compared to lightly populated areas, the highly populated areas exhibited higher $NO_2$ concentration levels, primarily due to increased emissions and a more developed economy (Qiu et al., 2023). Among all subregions, the highest $NO_2$ concentrations for highly populated and supremely highly populated areas were found in the NC region, while the highest $NO_2$ concentrations for lightly populated areas were observed in the EC region. In the highly populated areas in the NC region, $NO_2$ concentrations

exceeded IT-2 levels and were nearly double the concentrations of lightly populated areas. NO$_2$ concentrations in highly populated areas of NWC, NC, CC, SWC, and SC exceeded the IT-3 levels. Only NC, CC, and EC exceeded the IT-3 level for moderately populated areas. Furthermore, all the subregions and their urbanization categories, including the lightly populated areas, depicted their NO$_2$ values higher than the AQG level.

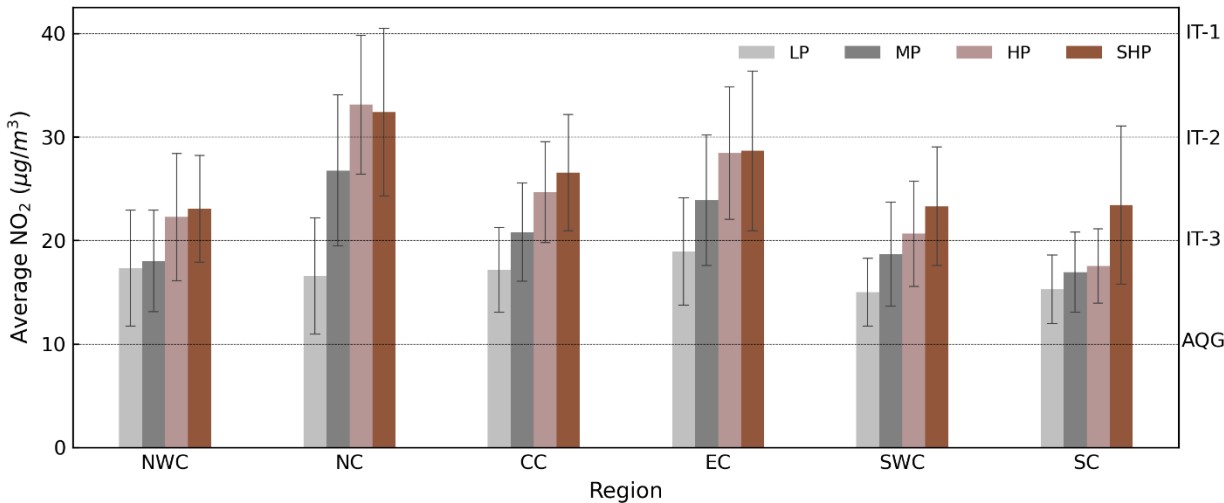

**Figure 10: Annual mean ground-level NO₂ concentrations for 2021 in subregions with different geolocations (e.g., NWC, NC, CC, EC, SWC, and SC) and urbanization levels (e.g., LP, MP, HP, and SHP). The vertical bars represent one sigma standard deviation.**

### 3.3 Seasonal variations of ground-level NO₂ concentrations

Similar to the annual average, the estimation of seasonal average NO$_2$ incorporated correction factors to address the data missing issues resulting from clouds and in the nighttime. Based on the bias-corrected NO$_2$ data, the seasonal averages NO$_2$ concentrations for lightly populated, moderately populated, highly populated, and supremely highly populated areas are shown in Fig. 11. Among all subregions, the ground-level NO$_2$ concentrations were highest in winter. This can be attributed to the more stable atmospheric structure and lower precipitation during this season, which creates less favourable conditions for the dispersion and deposition of ground-level NO$_2$. Additionally, the reduced photolysis rate of NO$_2$ due to low temperatures in winter leads to an increased residence time of NO$_2$ in the atmosphere (Xu et al., 2021). The temperature inversion in winter can further prolong the lifetime of the ground-level NO$_2$, leading to higher accumulations near the ground. Furthermore, the elevated concentrations in winter can be attributed to increased energy consumption for heating purposes.

Among the six subregions, NC and EC depicted the highest NO$_2$ concentrations, reaching levels close to IT-1 (40 µg/m$^3$) in winter for highly populated areas. Conversely, the lowest ground-level NO$_2$ concentrations were observed during summer for all six subregions. During this season, the increased precipitation coupled with the monsoon-induced atmospheric convection fosters wet deposition and dispersion of ground-level NO$_2$. Additionally, abundant sunlight promotes the decomposition of NO$_2$. Furthermore, the NO$_2$ emissions are generally lower in summer than in winter (Bhattarai et al., 2021; Fan et al., 2020; Tian et al., 2019). Considering the different population densities in the

subregions, the NO₂ pollution levels were lowest in lightly populated areas and highest in highly populated areas for all seasons. In lightly populated areas, the average NO₂ concentrations were approximately 50 % of those observed in highly populated areas.

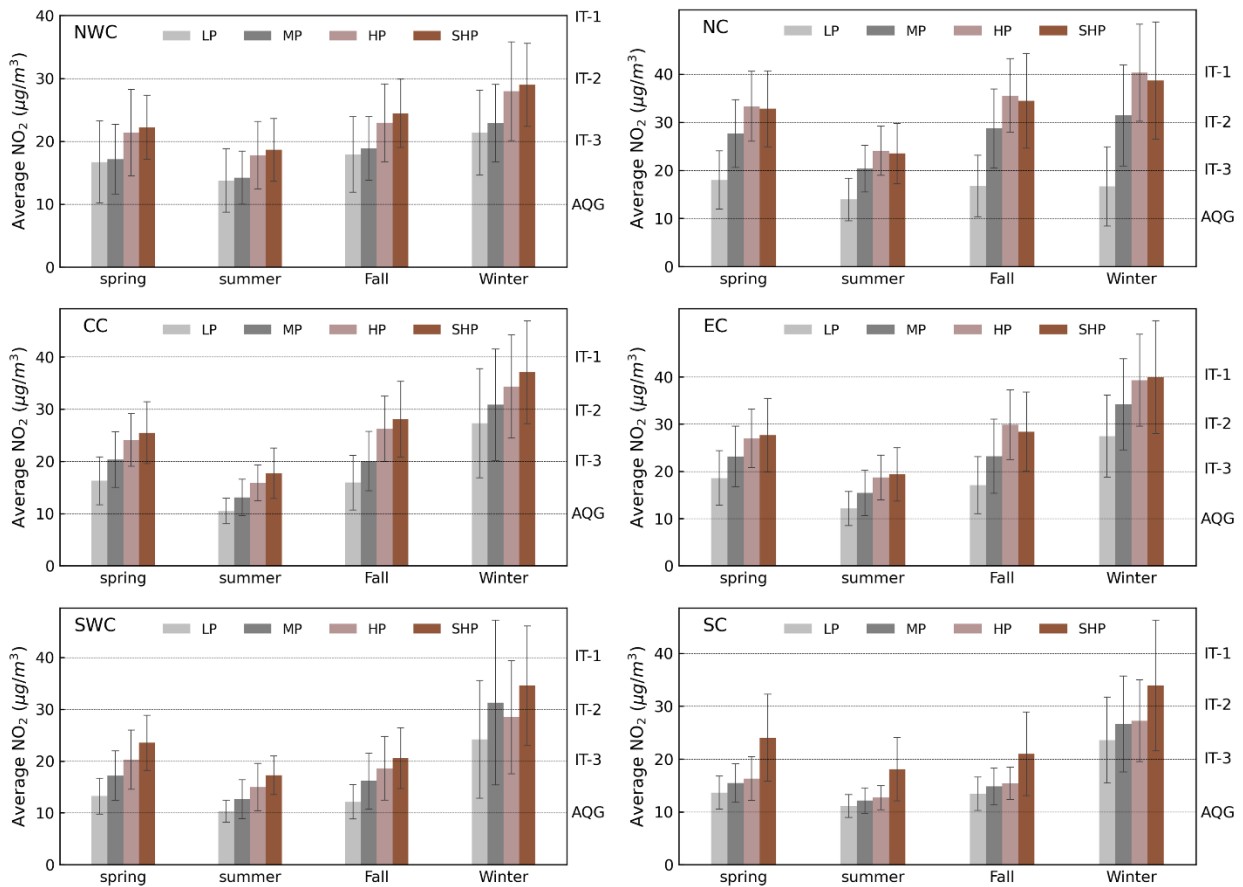

**Figure 11: Seasonal variations in ground-level NO₂ concentrations for 2021 in subregions with different geolocations (e.g., NWC, NC, CC, EC, SWC, and SC) and urbanization levels (e.g., LP, MP, HP, and SHP). The vertical bars represent one sigma standard deviation.**

### 3.4 Diurnal variations of ground-level NO₂ concentrations

The estimations of hourly averaged ground-level NO₂ concentrations incorporated correction factors to address data gaps caused by clouds. Based on the bias-corrected NO₂ data, Fig. 12 shows the spatial distribution of average ground-level NO₂ concentrations for each hour between 8:00 AM and 3:00 PM in 2021. Consistent spatial patterns were observed during this time range, with higher ground-level NO₂ concentrations in highly populated urban areas characterized by elevated NO$_x$ emissions. In the morning, clear indications of high ground-level NO₂ concentrations were noticed over urban centres, reflecting NO$_x$ emissions related to traffic. The spatial gradients of ground-level NO₂ concentrations were notably pronounced from urban centres to the outskirts during this time. However, these spatial gradients were less pronounced during noon and afternoon hours. Compared to the highly populated urban areas, ground-level NO₂ distributions in lightly populated areas displayed lower diurnal variability. These variations in

ground-level NO₂ distributions can be attributed to changes in NOₓ emission patterns, meteorological conditions, and photochemistry throughout different times of the day (Shen et al., 2023). For instance, Xu et al. (2023) observed the minimum NO₂ lifetime at noon, which can be attributed to higher photochemical reaction rates resulting from increased temperature and ultraviolet radiation (Gao et al., 2023).

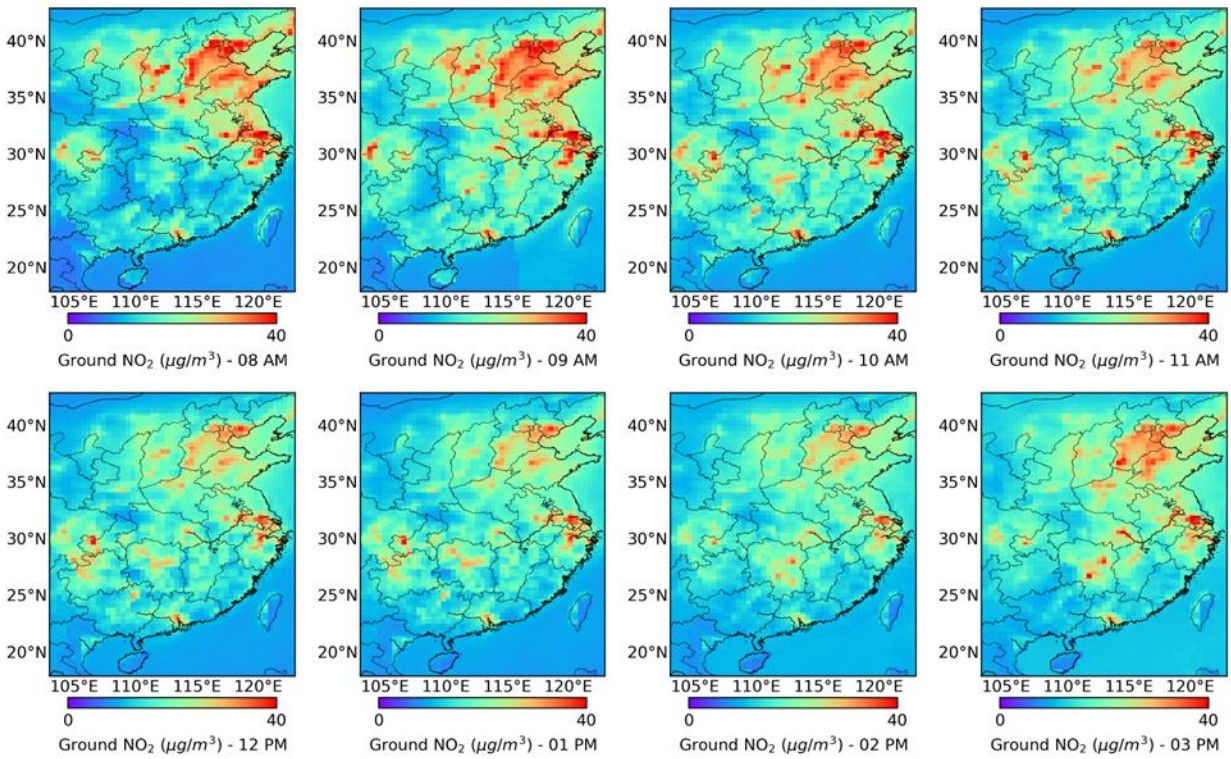

**Figure 12: Spatial distributions of the average ground-level NO₂ concentrations for each hour between 8:00 AM and 3:00 PM in 2021.**

The diurnal variations of ground-level NO₂ concentrations for the subregions are illustrated in Fig. 13. In most subregions, the peak of ground-level NO₂ was observed between 8:00 AM and 9:00 AM in highly populated areas. Additionally, a slight increase in NO₂ concentrations was observed in the late afternoon (i.e., 3:00 PM). In lightly populated and moderately populated areas, NWC and NC depicted a decreasing trend from 8:00 AM to 1:00 PM, followed by a slight increase at 2:00 PM and 3:00 PM. Lightly populated areas of CC showed an increasing trend from 8:00 AM to 10:00 AM, followed by a nearly constant value. However, moderately populated areas of CC showed a decreasing trend from 8:00 AM to 1:00 PM and then displayed an increasing trend at 2:00 PM and 3:00 PM. EC exhibited increasing values from 8:00 AM to 9:00 AM, followed by a decreasing trend until 2:00 PM, and again increased until 3:00 PM for both lightly populated and moderately populated areas. In lightly populated and moderately populated areas of SWC, NO₂ concentrations showed an increasing trend from 8:00 AM to 10:00 AM, followed by a decreasing trend throughout the afternoon. For the SC region, NO₂ concentrations remained relatively consistent from 8:00 AM to 10:00 AM, followed by a decreasing trend in both lightly populated and moderately populated areas.

Overall, highly populated areas exhibited peak ground-level $NO_2$ concentrations during the early morning rush hours (8:00 AM - 9:00 AM), followed by a decreasing trend. The minimum $NO_2$ levels were observed at 1:00 PM - 2:00 PM, with a slight increase observed at 3:00 PM. This diurnal pattern of ground-level $NO_2$ concentrations aligns with the findings of Zhang et al. (2023). The decrease in $NO_2$ levels from early morning to afternoon can be attributed to reduced traffic emissions, increased photochemical consumption, and higher NMH levels (Ahmad et al., 2024; Xie et al., 2016). In lightly populated and moderately populated areas, a slight morning peak was observed around 9:00 AM or 10:00 AM, occurring later than the peak observed in urban areas. This delayed morning peak in these areas can be attributed to regional dispersions originating from urban sources. The diurnal pattern of ground-level $NO_2$ concentrations observed in this study is consistent with previous studies using ground-based air quality monitoring stations (Shen et al., 2023; Yu et al., 2020; Zhao et al., 2016).

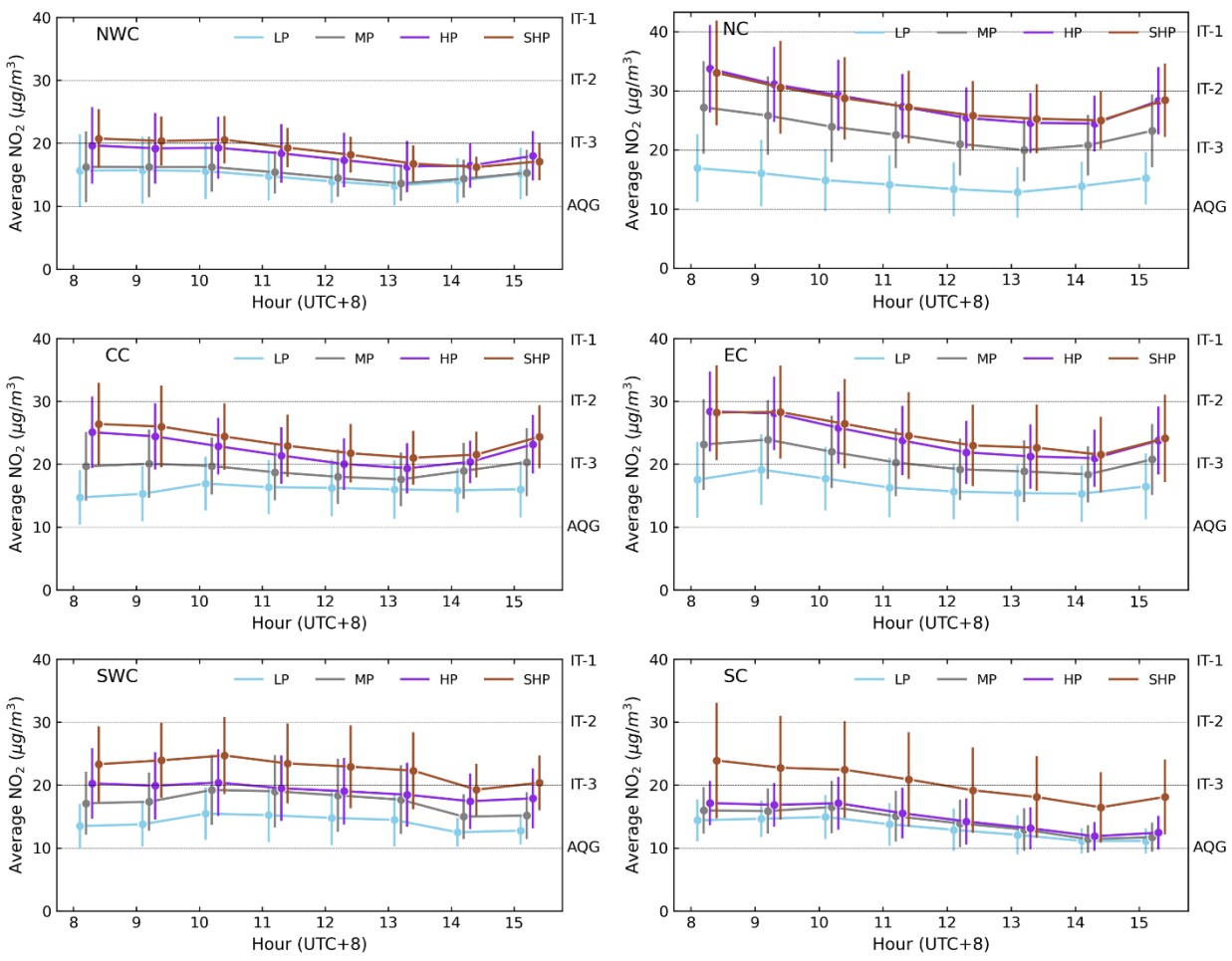

**Figure 13: Diurnal variations in ground-level $NO_2$ concentrations from 8:00 AM to 3:00 PM for 2021 in subregions with different geolocations (e.g., NWC, NC, CC, EC, SWC, and SC) and urbanization levels (e.g., LP, MP, HP, and SHP). The vertical bars represent one sigma standard deviation.**

**4 Discussion**

The scientific contributions of this study are summarized as follows. First, the results of this study have contributed to enriching our scientific understanding of the relationship between columnar $NO_2$ and ground-level $NO_2$. We have proven that the mixing height of $NO_2$ plays a key role in linking satellite-derived VCDs of $NO_2$ with ground-level concentrations, though the impacts of NMH were rarely considered in a direct manner in previous studies. Secondly, the analyses in this study have improved our understanding of the spatiotemporal variations of $NO_2$, particularly the diurnal variations that cannot be obtained from common polar-orbiting satellite measurements. The diurnal variations in $NO_2$ concentration differ between urban and rural areas, resulting from the different emission sources and pollutant dispersion characteristics. Thirdly, the analyses of $NO_2$ variation have policy implications for air pollution control. It was found that the spatial coincidence between $NO_2$ concentrations and population density increased overall population exposure and the associated health impacts. This suggests that for more effective reduction of overall population exposure and better protection of public health, control efforts should be further targeted at highly populated and highly polluted areas. Additionally, land-use and city planning should encourage population redistribution away from the most heavily polluted regions.

PBL characteristics are pivotal in regulating the vertical dispersion and horizontal transport of atmospheric pollutants, subsequently determining the vertical variations of $NO_2$ and its concentration at the Earth's surface (Akther et al., 2023; Xiang et al., 2019). Results in this study highlight the key role of the mixing height of $NO_2$ in linking satellite-derived VCDs of $NO_2$ with ground-level concentrations. To convert the VCDs of $NO_2$ into ground-level $NO_2$ concentrations, previous conversion models have used PBLH as a proxy of the NMH, because of its ability to regulate ground-level pollution levels. For example, within a stable PBL, pollutants like $NO_2$ from ground sources mainly accumulate near the ground surface (Levi et al., 2020). Intense solar heating can induce elevated temperatures, fostering an unstable PBL that is conducive to the upward dispersion of air pollutants including $NO_2$ (Kalmus et al., 2022; Su et al., 2020). The wind pattern is connected to atmospheric stability and can impact $NO_2$ levels by modifying pollutants' dispersion and horizontal transport (Yin et al., 2019). High surface air pressure often leads to large-scale sinking air motion, resulting in the limited vertical diffusion of $NO_2$ (Chow et al., 2018). Elevated relative humidity levels act as a suppressive factor, constraining the PBLH and exacerbating the accumulation of pollutants near the ground (Xiang et al., 2019). Therefore, different meteorological factors significantly impact the vertical distribution of $NO_2$ in the atmosphere (Huang et al., 2021). This study developed a conversion model that directly considers the impacts of the NMH. The predictions of NMH from the inner model directly incorporated the impacts of meteorological parameters (T, P, WS, RH, DP, VIS, and PRECIP). It was found that temperature, wind speed, dew point, and visibility were positively correlated with NMH, while relative humidity and air pressure mainly demonstrated an inverse relationship (Ahmad et al., 2024). The atmosphere's dynamic and thermodynamic aspects played crucial roles in developing the vertical structure of $NO_2$. The incorporation of the NMH in the model paved the way to refine the processes of converting satellite-derived columnar measurements into ground-level $NO_2$ concentrations.

Two models were tested and trained: Model I, which did not consider NMH, and a nested Model II, which incorporated NMH. The validation results demonstrated that nested Model II exhibited more promising outcomes than Model I, suggesting that including NMH significantly influenced the model's performance. Including NMH as an input

parameter in the machine learning model could better capture the vertical distributions of $NO_2$ and thus predict ground-level $NO_2$ concentrations with improved accuracy and performance. Additionally, the hour-by-hour 10-fold cross-validation depicted a distinct improvement in the ground-level $NO_2$ estimations for nested Model II considering NMH as an input parameter (Fig. S5 for Model I without NMH and Fig. S6 for nested Model II with NMH). The $R^2$ values for Model I without NMH were 0.63 for 8:00 AM, 0.70 for 9:00 AM, 0.69 for 10:00 AM to 1:00 PM, 0.55 for 2:00 PM, and 0.39 for 3:00 PM. The improved $R^2$ values for nested Model II, which includes NMH, were 0.85 for 8:00 AM, 0.90 for 9:00 to 11:00 AM, 0.91 for 12:00 PM, 0.93 for 1:00 PM, 0.89 for 2:00 PM, and 0.85 for 3:00 PM. Similarly, nested Model II, considering the NMH, depicted significantly reduced biases compared to Model I without NMH. The ground-level $NO_2$ estimations for all hours were significantly improved when considering NMH, as it directly incorporates the vertical distributions of $NO_2$. During the early morning hours, most of the $NO_2$ is distributed near the ground. However, as the day progresses, NMH increases, and the ground-level $NO_2$ tends to be mixed vertically. Further, the improvements in ground-level $NO_2$ estimations were assessed using 10-fold cross-validation for different population categories, i.e., lightly populated, moderately populated, highly populated, and supremely highly populated. The nested Model II, considering NMH, depicted notable improvements compared to Model I without NMH (Fig. S7). The improved $R^2$ values for nested Model II considering NMH were 0.91 for lightly populated areas and 0.92 for the other three population categories compared to Model I without NMH, which depicted an $R^2$ value of 0.63 for lightly populated, 0.73 for moderately populated, 0.77 for highly populated, and 0.74 for supremely highly populated areas. The RMSE for nested Model II considering NMH was improved and observed below 5 $\mu g/m^3$ for all population categories compared to Model I without NMH, which depicted RMSE values around 8-9 $\mu g/m^3$ for different population categories. The MAPE for nested Model II considering NMH was also improved for all population categories, and around 15 % and lower values were observed. These improvements depict that nested Model II considering NMH effectively captures the spatial distributions of vertical mixing of ground-level $NO_2$ across all population categories. The spatiotemporal distributions and diurnal patterns of NMH are previously described by Ahmad et al. (2024). Compared to Model I without NMH, the performance of the ground-level $NO_2$ estimations through nested Model II considering NMH showed significant improvement at the grid points where ground-based observations were available (Fig. S8). The correlation coefficients for grid-based 10-fold cross-validation were improved to 0.8-1.0 for nested Model II considering NMH compared to Model I without NMH, which depicted lower correlation coefficients. Furthermore, nested Model II considering NMH also depicted lower RMSE values for grid-based estimations.

GEMS, the world's first GEO-based environmental satellite instrument, offers a new opportunity for monitoring air quality across extensive regions, providing unprecedented spatial and temporal resolution. The quality of GEMS $NO_2$ VCDs, obtained from the level 2 product, has been evaluated using ground-based instruments in various regions. Encouragingly, a good agreement has been observed between the GEMS $NO_2$ VCDs and measurements from various ground-based instruments (Ahmad et al., 2024; Kim et al., 2023; Li et al., 2023). The results presented in this study emphasize the significant advantage of geostationary satellites in providing air pollution information at an hourly resolution. They enable the assessment of diurnal variations in air pollution across different areas, ranging from lightly populated to supremely highly populated regions. This represents a substantial improvement over traditional LEO-

based satellite instruments. Furthermore, these GEO-based measurements are valuable supplements to traditional measurements from ground-based air quality monitoring networks, primarily concentrated in urban areas, leaving vast rural regions without observations.

The diurnal variations of ground-level $NO_2$ concentrations across China depicted distinct gradients across all subregions and population categories. This gradient reflects regional disparities in industrialization, urbanization, and transportation infrastructure of Chinese megacities and rural areas. Highly populated areas depicted the highest concentrations of ground-level $NO_2$ during the early morning hours, attributed to intensified vehicular traffic in the early morning hours and higher industrial emissions. In contrast, lightly populated areas exhibited lower ground-level $NO_2$ concentrations and a delayed peak of around one to two hours, indicating lesser anthropogenic influence and more contribution from regional transport contributed by the $NO_2$ emissions from highly populated areas. Various driving factors influence these diurnal variations in ground-level $NO_2$ concentrations, each contributing differently across different regions. For instance, anthropogenic emissions dominate in highly populated urban and suburban areas, characterized by traffic emissions peaking in the morning and late afternoon (Liu et al., 2018; Naiudomthum et al., 2022). This phenomenon is particularly pronounced in highly populated areas with high traffic density. As morning rush hours subside, reduced vehicular traffic activities in highly populated areas lead to a gradual decline in $NO_2$ emissions. However, atmospheric processes such as higher mixing height of $NO_2$, more dispersion, and dilution also come into play, resulting in reduced ground-level $NO_2$ concentrations. Increased turbulent mixing in the lower atmosphere helps disperse pollutants from their sources in highly populated areas, gradually decreasing ground-level $NO_2$ concentrations. Additionally, photochemistry also influences the diurnal variations of $NO_2$ concentrations. The ratio of $NO_2$ to $NO$ is influenced by radiation, ozone, and peroxyl radicals. During the daytime, $NO_x$ undergoes oxidation through radical-mediated reactions, forming nitric acid and organic nitrates, with their levels depending on radiation, ozone, and volatile organic compounds. As a result, the lifetime of $NO_2$ reaches its lowest point around noon, typically lasting a few hours during summer. Furthermore, atmospheric transport contributes to the diurnal variation of $NO_2$, particularly in highly populated areas and their surrounding regions (Zhang et al., 2023). The hourly ground-level $NO_2$ concentration results presented in this study provide high-resolution information on the diurnal variations in ground-level $NO_2$ pollution levels across different regions and demographic patterns.

The spatial distribution of ground-level $NO_2$ concentrations in the study region revealed significant regional disparities, with higher levels observed in urban agglomerations with high population densities (e.g., BTH, YRD, and PRD regions) than in lightly populated areas (e.g., western China). Even within the NC region, the highly populated urban areas had $NO_2$ concentrations nearly double those of lightly populated rural areas. These spatial disparities are due to distributions of $NO_2$ emission sources that vary with population densities, decreasing from highly populated to lightly populated areas. In highly populated urban areas in regions like BTH, YRD, and PRD, mobile $NO_x$ emissions from dense road networks contribute to pronounced increase in $NO_2$ levels. Moreover, the short lifespan of $NO_2$ due to atmospheric chemical reactions results in elevated concentrations near emission sources in highly populated areas, such as roadways, accompanied by rapid declines in $NO_2$ concentrations with increasing distance from highly populated areas (Lee et al., 2018). Furthermore, the diverse terrains, land cover, and climates observed in subregions

with different population categories collectively influence vertical and horizontal airflows, rates of $NO_2$ formation and deposition, and contribute to spatiotemporal variations in ground-level $NO_2$ concentrations between the highly populated and lightly populated areas across China. Additionally, the population-weighted mean $NO_2$ concentrations were consistently higher than the spatial mean $NO_2$ concentrations in most provinces across China. This is due to the spatial coincidence between $NO_2$ concentrations and population density. These results indicate that the use of simple

spatial average concentrations can lead to a systematic underestimation of overall population exposure and the associated health impacts. It is important to use high-resolution $NO_2$ data to accurately quantify true population exposure. Furthermore, the adverse impacts of high $NO_2$ concentrations in highly populated urban areas suggest that for more effective reduction of overall population exposure and better protection of public health, control efforts should be further targeted at highly populated and highly polluted areas. Targeted control programs to reduce pollutant

levels at population hotspots should be more cost-effective than trying to reduce pollutant concentrations everywhere. Additionally, control policies can be implemented by encouraging the public to relocate to less polluted areas through land-use development and urban planning.

The GEMS measurements, while valuable, are subject to uncertainties and limitations. One of the primary challenges is the impact of cloudy conditions, which can affect the reliability of GEMS measurements. To address this issue, data

with a cloud fraction exceeding 30 % were intentionally excluded from the analysis. This approach aimed to strike a balance between obtaining an adequate number of measurements and minimizing the influence of cloud-contaminated data. Additionally, data with a solar zenith angle exceeding 70° were excluded. Regions with a higher likelihood of cloud cover had more missing data, and there was a relatively small sample size available in the early morning due to the absence of solar radiation. Another inherent limitation of satellite measurements is the lack of data during nighttime.

The lack of nighttime data and cloudy conditions leads to skewness in the GEMS measurements, especially for phenomena that exhibit diurnal variations. To align the satellite-estimated $NO_2$ with ground-measured $NO_2$, correction factors were applied for hourly, seasonal, and annual averages (see Sec. 2.7). These correction factors are based solely on the ground $NO_2$ measurements, which results in reduced and minimized biases associated with them. However, some limitations still exist, as these correction factors rely on an ancillary data source with low spatial resolution.

Spatially, the spatial distributions of the correction factors were obtained by interpolating the ground monitoring data. We made the assumption that the correction factors vary smoothly in the areas between different stations. However, atmospheric conditions and $NO_2$ emissions can vary significantly across different regions at different times of the day. Additionally, we applied a constant correction factor for seasonal and annual averages, which may not be able to correct the detailed bias from hour to hour. It is important to note that the data used in this study corresponds to version

1 of the GEMS product. Ongoing efforts are being made to enhance the accuracy of GEMS products, and subsequent versions are expected to offer improved quality and reliability.

Further, to explore the impact of missing GEMS $NO_2$ VCDs and associated biases on estimating average ground-level $NO_2$ concentrations between 8:00 AM and 3:00 PM, we calculated the difference between the average $NO_2$ concentrations derived from all ground measurements and the average ground-measured $NO_2$ concentrations when

satellite data was available. The hourly variations of these concentration differences for 2021 are presented in Fig. 14.

The issue of missing data consistently underestimated the average $NO_2$ concentrations for each hour. The degree of underestimation was higher during hours with more missing data. For instance, at 3:00 PM, 2:00 PM, 1:00 PM, and 8:00 AM, the mean underestimation was -6.27±2.38 $\mu g/m^3$, -4.38±1.94 $\mu g/m^3$, -2.60±2.50 $\mu g/m^3$, and -1.57±1.19 $\mu g/m^3$, respectively. The underestimation gradually decreased for 12:00 PM, 11:00 AM, and 9:00 AM. Notably, the underestimation was at its minimum for 10:00 AM, with a value of -0.16±1.61 $\mu g/m^3$.

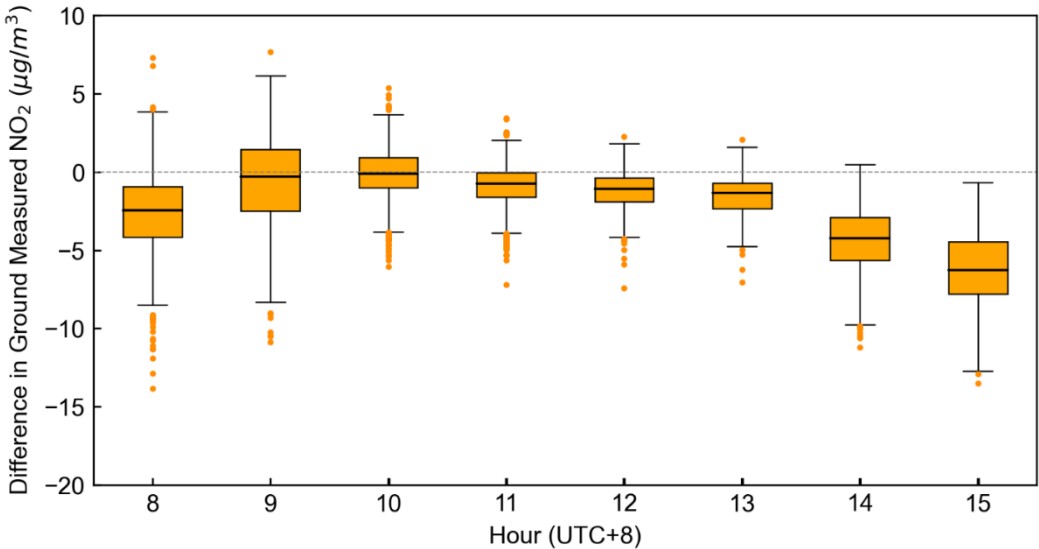

**Figure 14: Difference between the average $NO_2$ concentrations from all ground measurements and the average ground-measured $NO_2$ concentration when satellite data was available for each hour from 8:00 AM to 3:00 PM. The vertical bars represent whiskers that extend to the most extreme data points within 1.5 times the interquartile range from quartile 1 (25th percentile of data) and quartile 3 (75th percentile of the data).**

## 5 Conclusion

This study developed a nested machine learning model to incorporate the NMH as an input parameter in the methodological framework. The model's performance in predicting ground-level $NO_2$ concentrations from satellite columnar measurements was then explored. Among the testing and training of the two models, the model that considered the NMH as one of the input parameters demonstrated more promising results. This suggests that the inclusion of the NMH significantly impacts the model's performance. Furthermore, the NMH was identified as the second most important predictor variable after the GEMS $NO_2$ VCDs. The diurnal variations of satellite-derived ground-level $NO_2$ concentrations exhibited a clear gradient across all subregions, ranging from highly populated to lightly populated areas. In highly populated areas, peak ground-level $NO_2$ concentrations were observed during the early morning rush hours (8:00 AM - 9:00 AM). In areas categorized as lightly populated or moderately populated, a slight morning peak was observed around 9:00 AM or 10:00 AM, occurring later than in urban sites. In highly and supremely highly populated areas in northern China, $NO_2$ concentrations still exceeded the WHO IT-2 standards and were double the levels observed in lightly populated regions. These satellite-derived ground-level $NO_2$ concentrations provided high-resolution information on the diurnal variations of $NO_2$ pollution levels across different regions and

levels of urbanization. It is important to note that the GEMS measurements, while valuable, are subject to uncertainties and limitations, particularly due to the impact of cloudy conditions and the absence of nighttime data. Correction factors were applied in this study to mitigate these issues and address the inherent challenges of satellite measurements. Some limitations still exist, as these correction factors rely on an ancillary data source with low spatial resolution. Additionally, we applied a constant correction factor for seasonal and annual averages, which may not be able to

correct the detailed bias that occurs from hour to hour. Overall, the findings of this study enhance our understanding of the effects of the mixing height of $NO_2$ on the conversion of satellite-based columnar measurements to ground-level $NO_2$ concentrations. They also provide valuable insights into the spatial and diurnal patterns of ground-level $NO_2$ across China.

**Acknowledgements**

This work was supported by the NSFC/RGC Joint Research Project (Grant Nos. 42161160329 and N_HKUST609/21), the Research Grants Council of Hong Kong (Project Nos. GRF 16202120 and 16302220), and Laboratory of Optical Monitoring of Atmospheric Environment of HKUST(GZ).

**Data availability**

We thank the National Institute of Environmental Research (NIER) of South Korea for providing the GEMS data

(https://nesc.nier.go.kr). We thank the Institute for the Environment (IENV) and Environmental Central Facility (ENVF) of the Hong Kong University of Science and Technology (HKUST) for providing atmospheric and environmental data (http://envf.ust.hk/dataview/). Data are available upon requests.

**Author Contribution**

Lin CQ designed the analyses and Ahmad N carried them out. Lau AKH supervised the study. Kim J provided the

data. Yu FQ, TS Zhang, and Li CC performed the simulations. Li Y, Fung JCH, and Lao XQ edited the manuscript. Ahmad N and Lin CQ prepared the manuscript with contributions from all co-authors.

**Competing interests**

The authors declare that they have no conflict of interest. Further, one co-author is a member of the editorial board of ACP.

**Supplements**

It includes Figs. S1-S8

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

surface NO2 for Beijing reconstructed from surface data and satellite retrievals. Science of The Total Environment 904, 166693.