# Peer review of "Estimation of Ground-Level NO2 and its Spatiotemporal Variations in China Using GEMS Measurements and a Nested Machine Learning Model"

_EGUsphere, 2024_

## Author Response (AR1)

Dear Editor and Reviewers,

We are grateful to the reviewers for their helpful comments. We have made the modifications in response to their comments. Attached is a point-by-point response to the comments. We hope that you and the referees will find the changes satisfactory, and we look forward to hearing from you soon.

**RC1: 'Comment on egusphere-2024-558', Referee #1**

This study leverages advanced satellite measurements and machine learning techniques to estimate ground-level NO2 concentrations in China. The use of the GEMS measurements combined with a nested machine learning model marks an advanced approach to addressing the challenge of translating satellite-derived VCDs of NO2 into actionable ground-level concentration data. Incorporating the NMH into the prediction model not only demonstrates a methodological advancement but also highlights the crucial role of meteorological conditions in the dispersion of atmospheric pollutants. As the study achieves remarkable accuracy and provides comprehensive analyses of NO2 distribution patterns, I recommend the publication of this paper for Atmospheric Chemistry and Physics after minor revisions.

Specific Comments:

**Comment 1:**

The planetary boundary layer (PBL), represented as NMH in this study, is identified as a significant factor influencing the conversion of VCDs of NO2 to ground-level concentrations. Due to its importance as illustrated in Figure 5, there should be more discussions on the relationship between PBL and surface air pollution. I also suggest the authors acknowledge the previous study investigating the Relationships between the PBL and surface pollutants over China, as well as the influencing factors.

Response: Thank you for your valuable comments. We have added a new paragraph in the Introduction section (lines 89-103) and a new paragraph in the Discussion section (lines 485-505) to provide more elaboration on the impacts of the PBL on air pollution.

"Numerous past studies have highlighted the importance of the boundary layer structure in governing the occurrence and evolution of extreme air pollution episodes (Shi et al., 2020). A significant relationship between a surge in surface air pollutant concentrations and a shallow PBLH has been extensively reported (Miao et al., 2019; Su et al., 2020). It has also been recognized that air pollutants aloft can play a core role in the evolution of surface extreme pollution episodes via vertical mixing (Zhang and Rao, 1999). When the top of the mixing layer reaches the aloft pollutant-rich layer during the daytime, air pollutants can be entrained downwards, which rapidly increases surface air pollutant concentrations (Zhang et al., 2016). In addition to the vertical exchange, radiative absorption and scattering by pollutants can modify the boundary layer structure and consequently affect ground-level pollutant concentrations. For instance, high loadings of scattering pollutants can cool the air near the ground and result in a more stable boundary layer, which further worsens air quality (Li et al., 2017). As a result, the PBLH has been used as a proxy of the NMH because of its ability to regulate near-surface pollution levels. However, as $NO_2$ may not be uniformly distributed within the planetary boundary layer, a significant difference may exist between the PBLH and NMH. It is important to develop a conversion model that directly considers the impacts of the NMH, which paves the way to refine the processes of converting satellite-derived columnar measurements into ground-level $NO_2$ concentrations (Ahmad et al., 2024)."

"PBL characteristics are pivotal in regulating the vertical dispersion and horizontal transport of atmospheric pollutants, subsequently determining the vertical variations of $NO_2$ and its concentration at the Earth's surface (Akther et al., 2023; Xiang et al., 2019). Results in this study highlight the key role of the mixing height of $NO_2$ in linking satellite-derived VCDs of $NO_2$ with ground-level concentrations. To convert the VCDs of $NO_2$ into ground-level $NO_2$ concentrations, previous conversion models have used PBLH as a proxy of the NMH, because of its ability to regulate ground-level pollution levels. For example, within a stable PBL, pollutants like $NO_2$ from ground sources mainly accumulate near the

ground surface (Levi et al., 2020). Intense solar heating can induce elevated temperatures, fostering an unstable PBL that is conducive to the upward dispersion of air pollutants including $NO_2$ (Kalmus et al., 2022; Su et al., 2020). The wind pattern is connected to atmospheric stability and can impact $NO_2$ levels by modifying pollutants' dispersion and horizontal transport (Yin et al., 2019). High surface air pressure often leads to large-scale sinking air motion, resulting in the limited vertical diffusion of $NO_2$ (Chow et al., 2018). Elevated relative humidity levels act as a suppressive factor, constraining the PBLH and exacerbating the accumulation of pollutants near the ground (Xiang et al., 2019). Therefore, different meteorological factors significantly impact the vertical distribution of $NO_2$ in the atmosphere (Huang et al., 2021). This study developed a conversion model that directly considers the impacts of the NMH. The predictions of NMH from the inner model directly incorporated the impacts of meteorological parameters (T, P, WS, RH, DP, VIS, and PRECIP). It was found that temperature, wind speed, dew point, and visibility were positively correlated with NMH, while relative humidity and air pressure mainly demonstrated an inverse relationship (Ahmad et al., 2024). The atmosphere's dynamic and thermodynamic aspects played crucial roles in developing the vertical structure of $NO_2$. The incorporation of the NMH in the model paved the way to refine the processes of converting satellite-derived columnar measurements into ground-level $NO_2$ concentrations."

**Comment 2:**

Section 2.6 is the key section for this paper, since it present the details of machine learning model for this study. While the nested machine learning model demonstrates superior performance in estimating ground-level NO2 concentrations, the methodology section could benefit from a more clear discussion of the advantage of XGBoost regression model, as well as feature selection process, and the rationale behind choosing specific meteorological parameters as predictors.

Response: Thanks for your comments. The XGBoost algorithm has proven to be useful in various air quality studies, including those focusing on the conversion between satellite-based column measurements and ground-level concentrations (Shao et al., 2023; Zhao et al., 2023). We have added a new paragraph to discuss the advantages of the XGBoost regression model in lines 197-214.

"XGBoost stands out as a notably efficient end-to-end gradient boosting tree framework, adept at transforming numerous weak learners into robust ones through boosting. This framework frequently demonstrates reduced computational overhead and enhanced predictive accuracy when compared with alternative ensemble tree models (Chen and Guestrin, 2016). Moreover, XGBoost exhibits a lower susceptibility to overfitting by mitigating the bias within the context of bias-variance decomposition. XGBoost has been empirically demonstrated to adeptly capture nonlinear relationships between predictions and predictors, yielding precise estimations through its regularized boosting methodology. This approach constructs the ultimate model by iteratively refining simpler and weaker models, each subsequent tree learning from its predecessors and updating residual errors via gradient descent to optimize the loss function. Within the XGBoost framework, an augmented penalty term is incorporated into the error function to fine-tune the objective function, thereby smoothing the final learned weights and mitigating overfitting tendencies. Additionally, to further mitigate overfitting, feature sub-sampling and shrinkage techniques are integrated (Liu 2021). The study by Van et al. (2022) also demonstrated the XGBoost algorithm as the most suitable lightweight algorithm based on the comparative analysis of three machine learning models, i.e., XGBoost, Decision Tree, and Random Forest. The XGBoost algorithm has proven to be useful in various air quality studies, including those focusing on the conversion between satellite-based column measurements and ground-level concentrations (Shao et al., 2023; Zhao et al., 2023). More details on the XGBoost regression model can be found in Chi et al. (2022). The XGBoost model was implemented in this study to convert columnar measurements into ground-level $NO_2$ concentrations."

All common meteorological variables that are available from the ground monitoring network were used in this study. Therefore, we did not choose specific meteorological factors. The abilities of these meteorological variables

to regulate near-surface $NO_2$ levels are ranked as the feature importance in the XGBoost regression model. We clarified the use of meteorological variables in lines 225-232:

> "All common meteorological variables available from the ground monitoring network were used in this study. The ability of these meteorological variables to regulate near-surface $NO_2$ levels is ranked by feature importance in the XGBoost regression model. In our previous study, these meteorological parameters were shown to impact the vertical mixing of $NO_2$ to varying extents (Ahmad et al., 2024). For instance, elevated temperatures are conducive to the upward mixing of air pollutants. Increased wind speed is associated with an unstable atmosphere and can impact $NO_2$ levels by modifying the vertical dispersion and horizontal transport of air pollutants. Increased surface air pressure often leads to large-scale sinking air motion, which suppresses the vertical dispersion of $NO_2$."

**Comment 3:**

The study mentions the challenges posed by cloudy conditions and the lack of nighttime data in interpreting GEMS measurements. While correction factors were applied to mitigate these issues, a more detailed discussion on the limitations and potential biases introduced by these factors would be beneficial. This discussion of limitations can be also included or mentioned in the conclusion section.

Response: Thank you very much for your comments. The correction factors were applied to mitigate the issues of missing data. We created a new section (Section 2.7) to summarize the calculation of correction factors. The correction factors are only based on the ground $NO_2$ measurements, which results in reduced and minimized biases associated with them. However, some limitations still exist, as these correction factors rely on an ancillary data source with a low spatial resolution. Spatially, the spatial distributions of the correction factors were obtained from interpolation of the ground monitoring data. We made the assumption that the correction factors vary smoothly in the areas between different stations. However, the atmospheric conditions and $NO_2$ emissions can vary significantly across different regions at different times of the day. Additionally, we applied a constant correction factor for seasonal and annual averages, which may not be able to correct the detailed bias from hour to hour. We add the limitations for the correction factors in lines 602-611:

> "The lack of nighttime data and cloudy conditions leads to skewness in the GEMS measurements, especially for phenomena that exhibit diurnal variations. To align the satellite-estimated $NO_2$ with ground-measured $NO_2$, correction factors were applied for hourly, seasonal, and annual averages (see Sec. 2.7). These correction factors are based solely on the ground $NO_2$ measurements, which results in reduced and minimized biases associated with them. However, some limitations still exist, as these correction factors rely on an ancillary data source with low spatial resolution. Spatially, the spatial distributions of the correction factors were obtained by interpolating the ground monitoring data. We made the assumption that the correction factors vary smoothly in the areas between different stations. However, atmospheric conditions and $NO_2$ emissions can vary significantly across different regions at different times of the day. Additionally, we applied a constant correction factor for seasonal and annual averages, which may not be able to correct the detailed bias from hour to hour."

We also included this limitation in the conclusion (lines 645-647):

> "Some limitations still exist, as these correction factors rely on an ancillary data source with low spatial resolution. Additionally, we applied a constant correction factor for seasonal and annual averages, which may not be able to correct the detailed bias that occurs from hour to hour."

**RC2: 'Comment on egusphere-2024-558', Referee #2**

Overview:

This paper introduced a machine learning model to estimate ground-level NO2 concentrations from geostationary satellite-derived NO2 vertical column densities (VCDs). The overall conclusions are that utilizing NO2 mixing

height (NMH) can improve the accuracy of ground-level NO2 concentration estimates, and that satellite-derived ground-level NO2 concentration presents a population-based gradient.

Although this manuscript provides a few pieces of information that I believe are suitable for publication, it is riddled with grammar and technical issues and requires major revisions. Extensive simple grammar corrections should not be on the peer reviewers to fix at this stage, and such issues did make it difficult to understand the authors' justification behind their conclusions. I also found the present document more like a technical report rather than a research paper, as plenty of scientific discussions are missing.

Major Comments:

**Comment 1:**

The weakest point in the manuscript is the discussion of the results. More than two-thirds of the 'Discussions' section repeats what have already been presented in the 'Results' section. The authors should expand more on the scientific principles underlying the results in the 'Discussions' section.

Response: Thanks for your valuable comments. We have made efforts to improve the grammar throughout the manuscript. Additionally, we have thoroughly revised the Discussion section. Specifically, we have added a paragraph at the beginning of the Discussion section to summarize the scientific contributions of this study (lines 472-484).

> "The scientific contributions of this study are summarized as follows. First, the results of this study have contributed to enriching our scientific understanding of the relationship between columnar $NO_2$ and ground-level $NO_2$. We have proven that the mixing height of $NO_2$ plays a key role in linking satellite-derived VCDs of $NO_2$ with ground-level concentrations, though the impacts of NMH were rarely considered in a direct manner in previous studies. Secondly, the analyses in this study have improved our understanding of the spatiotemporal variations of $NO_2$, particularly the diurnal variations that cannot be obtained from common polar-orbiting satellite measurements. The diurnal variations in $NO_2$ concentration differ between urban and rural areas, resulting from the different emission sources and pollutant dispersion characteristics. Thirdly, the analyses of $NO_2$ variation have policy implications for air pollution control. It was found that the spatial coincidence between $NO_2$ concentrations and population density increased overall population exposure and the associated health impacts. This suggests that for more effective reduction of overall population exposure and better protection of public health, control efforts should be further targeted at highly populated and highly polluted areas. Additionally, land-use and city planning should encourage population redistribution away from the most heavily polluted regions."

Following the comments from the reviewers, we have expanded the discussions on the scientific principles underlying the results, focusing on the three key aspects mentioned above. First, we discussed the impacts of the mixing height of $NO_2$ (lines 485-537):

[revised manuscript text omitted]

**Comment 2:**

The title and abstract indicate that this paper aims at improving ground-level NO2 estimation. However, the only figures that present such improvements are Figures 4 and 12. The manuscript also keeps talking about different patterns of ground-level NO2 concentration between highly and lightly populated areas. But how the improvements differ between these regions (and at different hours of the day)? How the estimates perform at the grid points where ground-based observations are available?

Response: Thank you very much for your comments. In this study, we aimed to develop a nested model to improve the estimation of ground-level $NO_2$ and enrich our understanding of the spatial and temporal variations in $NO_2$ concentration using measurements from new geostationary satellite. The title of the manuscript has been revised as:

"Estimation of Ground-Level $NO_2$ and its Spatiotemporal Variations in China Using GEMS Measurements and a Nested Machine Learning Model".

Additionally, the following part has been added to assess the improvements at different hours of the day, improvements between different regions and performance of estimates at grid points where ground-based observations are available (lines 506-537).

"Two models were tested and trained: Model I, which did not consider NMH, and a nested Model II, which incorporated NMH. The validation results demonstrated that nested Model II exhibited more promising outcomes than Model I, suggesting that including NMH significantly influenced the model's performance. Including NMH as an input parameter in the machine learning model could better capture the vertical distributions of $NO_2$ and thus predict ground-level $NO_2$ concentrations with improved accuracy and performance. Additionally, the hour-by-hour 10-fold cross-validation depicted a distinct improvement in the ground-level $NO_2$ estimations for nested Model II considering NMH as an input parameter (Fig. S5 for Model I without NMH and Fig. S6 for nested Model II with NMH). The $R^2$ values for Model I without NMH were 0.63 for 8:00 AM, 0.70 for 9:00 AM, 0.69 for 10:00 AM to 1:00 PM, 0.55 for 2:00 PM, and 0.39 for 3:00 PM. The improved $R^2$ values for nested Model II, which includes NMH, were 0.85 for 8:00 AM, 0.90 for 9:00 to 11:00 AM, 0.91 for 12:00 PM, 0.93 for 1:00 PM, 0.89 for 2:00 PM, and 0.85 for 3:00 PM. Similarly, nested Model II, considering the NMH, depicted significantly reduced biases compared to Model I without NMH. The ground-level $NO_2$ estimations for all hours were significantly improved when considering NMH, as it directly incorporates the vertical distributions of $NO_2$. During the early morning hours, most of the $NO_2$ is distributed near the ground. However, as the day progresses, NMH increases, and the ground-level $NO_2$ tends to be mixed vertically. Further, the improvements in ground-level $NO_2$ estimations were assessed using 10-fold cross-validation for different population categories, i.e., lightly populated, moderately populated, highly populated, and supremely highly populated. The nested Model II, considering NMH, depicted notable improvements compared to Model I without NMH (Fig. S7). The improved $R^2$ values for nested Model II considering NMH were 0.91 for lightly populated areas and 0.92 for the other three population categories compared to Model I without NMH, which depicted an $R^2$ value of 0.63 for lightly populated, 0.73 for moderately populated, 0.77 for highly populated, and 0.74 for supremely highly populated areas. The RMSE for nested Model II considering NMH was improved and observed below 5 $\mu g/m^3$ for all population categories

compared to Model I without NMH, which depicted RMSE values around 8-9 μg/m$^3$ for different population categories. The MAPE for nested Model II considering NMH was also improved for all population categories, and around 15 % and lower values were observed. These improvements depict that nested Model II considering NMH effectively captures the spatial distributions of vertical mixing of ground-level NO$_2$ across all population categories. The spatiotemporal distributions and diurnal patterns of NMH are previously described by Ahmad et al. (2024). Compared to Model I without NMH, the performance of the ground-level NO$_2$ estimations through nested Model II considering NMH showed significant improvement at the grid points where ground-based observations were available (Fig. S8). The correlation coefficients for grid-based 10-fold cross-validation were improved to 0.8-1.0 for nested Model II considering NMH compared to Model I without NMH, which depicted lower correlation coefficients. Furthermore, nested Model II considering NMH also depicted lower RMSE values for grid-based estimations."

**Minor Comments:**

**Comment:** Line 122: What is the nominal spatial resolution of GEMS NO2 product used in this study?

Response: Thanks for your comments. We clarified the spatial resolution of the GEMS data in lines 136-139:

"The nominal spatial resolution of the GEMS NO$_2$ product used in this study was 7 km × 7.7 km. Despite the irregular shape of satellite measurement pixels due to east-to-west scans, this study performed re-gridding, which standardized the VCDs of NO$_2$ onto a regular grid of 0.2° × 0.4° by calculating the average of all the NO$_2$ VCDs within the 0.2° × 0.4° grid from 8:00 AM to 3:00 PM local time in China."

**Comment:** Line 124: Please provide some information on how NO2 VCDs are standardized. Line 160 mentioned bi-linear interpolation, but it is for meteorological variables.

Response: Thank you very much for your comments. In this study, we standardized the VCDs of NO$_2$ onto a regular grid of 0.2° × 0.4° by calculating the average of all the NO$_2$ VCDs within the 0.2° × 0.4° grid. We clarified it in lines 136-139:

"Despite the irregular shape of satellite measurement pixels due to east-to-west scans, this study performed re-gridding, which standardized the VCDs of NO$_2$ onto a regular grid of 0.2° × 0.4° by calculating the average of all the NO$_2$ VCDs within the 0.2° × 0.4° grid from 8:00 AM to 3:00 PM local time in China."

**Comment:** Line 135: … divided the study region into four areas … -> … divided the study area into four categories …

Response: Thanks for your comments. We have revised it accordingly (lines 149-150).

"Based on population density, we divided the study region into four categories."

**Comment:** Line 253: How is the month of the year numbered exactly? If 1 to 12 is used for January to December, then cold months would be around 12 to 2, which may affect SHAP values shown in Figure 6.

Response: Thanks for your valuable comments. We used a common method to number the months. The months are numbered from 1 to 12, corresponding to January through December, exactly as per the real months of the observations. Using alternative numbering methods may increase the complexity. Additionally, the month variable has a relatively small contribution of only 3.23% to the model's performance. The month variable served mainly as an auxiliary factor, and the SHAP values were mostly clustered around zero. The major variables are GEMS NO$_2$ VCDs and NMH. We clarified the numbering of the month variable in lines 225-226:

"The months are numbered from 1 to 12, corresponding to January through December, exactly as per the real months of the observations."

**Comment:** Line 259: Figure 6 indicates that lower T corresponds to lower NO2. How does it relate to 'worsened' ground-level NO2 pollution? And your reasoning 'air stagnation' may be wrong here.

**Response:** Thanks for your comments. When the feature values of temperature are large, the SHAP value is positive and may have a positive impact on the ground-level $NO_2$ predictions, but the impact value is not large. Some values with smaller feature values also have a positive impact on the model. It is noted that the SHAP values for the meteorological variables, including temperature, are all small, clustered around zero, and have limited influence on the prediction results. The major and distinct impact on the model's performance for predicting ground-level $NO_2$ concentrations is observed for GEMS $NO_2$ VCDs and NMH. We have rewritten the paragraph to highlight our focus (lines 330-333):

> "However, it is noted that the SHAP values for the meteorological variables, including temperature, are all small, clustered around zero, and have limited influence on the prediction results. The major and distinct impact on the model's performance for predicting ground-level $NO_2$ concentrations is observed for GEMS $NO_2$ VCDs and NMH."

**Comment:** Line 260: Figure 6 does not indicate this pattern. Please either quantify the impact of RH and dew point explicitly or remove this sentence.

**Response:** Thanks for your comments. We have removed the sentence.

**Comment:** Line 265: In this and the following sections, are ground-level NO2 concentration from ground-based observations or satellite-based estimates? Please clarify.

**Response:** Thanks for your comment. In this and the following sections, ground-level $NO_2$ concentrations are from satellite-based estimates. We have clarified it in the manuscript (lines 338-339):

> "Based on the satellite-derived ground-level $NO_2$ concentrations (mentioned as ground-level $NO_2$ concentrations from hereon)".

**Comment:** Line 266: Since this paragraph is talking about Fig. S1, I would suggest presenting the figure in the main text. Also, as the correction factor is important to the results of this study, how it is calculated should be presented in the main text or as an appendix. Related to the computation of correction factor, what is the possible maxima of m? Is it up to 24 (hours of a day)?

**Response:** Thanks for your comments.

(1) Fig. S1 has been moved to the main text as Fig. 7.

(2) The calculation of the correction factor is moved to the main text as Section 2.7.

(3) For a specific hour, the maximum value of $m$ in Eq. 1 is 365 for one year. We clarified it in line 263-264.

> "For a specific hour, the maximum value of $m$ index in Eq. 1 is 365 for one year."

For the annual correction factor, the maximum value of $m$ in Eq. 1 is 8760 for one year. We clarified it in line 280-281.

> "For the annual correction factor, the maximum value of $m$ index in Eq. 1 is 8760 for one year."

**Comment:** Line 350: Since Fig. S6 is discussed here, considering presenting the figure in the main text.

**Response:** Thanks for your comments. The figure has been moved to the main text as Fig. 12.

**Comment:** Line 425: Are NO2 and NO really in chemical equilibrium in the real atmosphere?

**Response:** Thanks for your comments. We revised the sentence and removed this particular description.

**Comment:** Line 444: The reasoning given here is too general. Consider adding some details/analysis specific to your results.

Response: Thanks for your comments. We had added more discussions on the spatial disparities of NO2 concentration and its implication for air pollution management in lines 572-594:

"The spatial distribution of ground-level $NO_2$ concentrations in the study region revealed significant regional disparities, with higher levels observed in urban agglomerations with high population densities (e.g., BTH, YRD, and PRD regions) than in lightly populated areas (e.g., western China). Even within the NC region, the highly populated urban areas had $NO_2$ concentrations nearly double those of lightly populated rural areas. These spatial disparities are due to distributions of $NO_2$ emission sources that vary with population densities, decreasing from highly populated to lightly populated areas. In highly populated urban areas in regions like BTH, YRD, and PRD, mobile $NO_x$ emissions from dense road networks contribute to pronounced increase in $NO_2$ levels. Moreover, the short lifespan of $NO_2$ due to atmospheric chemical reactions results in elevated concentrations near emission sources in highly populated areas, such as roadways, accompanied by rapid declines in $NO_2$ concentrations with increasing distance from highly populated areas (Lee et al., 2018). Furthermore, the diverse terrains, land cover, and climates observed in subregions with different population categories collectively influence vertical and horizontal airflows, rates of $NO_2$ formation and deposition, and contribute to spatiotemporal variations in ground-level $NO_2$ concentrations between the highly populated and lightly populated areas across China. Additionally, the population-weighted mean $NO_2$ concentrations were consistently higher than the spatial mean $NO_2$ concentrations in most provinces across China. This is due to the spatial coincidence between $NO_2$ concentrations and population density. These results indicate that the use of simple spatial average concentrations can lead to a systematic underestimation of overall population exposure and the associated health impacts. It is important to use high-resolution $NO_2$ data to accurately quantify true population exposure. Furthermore, the adverse impacts of high $NO_2$ concentrations in highly populated urban areas suggest that for more effective reduction of overall population exposure and better protection of public health, control efforts should be further targeted at highly populated and highly polluted areas. Targeted control programs to reduce pollutant levels at population hotspots should be more cost-effective than trying to reduce pollutant concentrations everywhere. Additionally, control policies can be implemented by encouraging the public to relocate to less polluted areas through land-use development and urban planning."

**Comment:** Line 470: The wording and the order of the sentence starting with 'The average ground-measured NO2 concentrations' is confusing, please revise.

Response: Thanks for your comments. We revised the sentence from "The average ground-measured $NO_2$ concentrations, when satellite data was available, consistently underestimated the average $NO_2$ concentrations from all ground measurements for each hour." to (line 618):

"The issue of missing data consistently underestimated the average $NO_2$ concentrations for each hour."

**Comment:** Figure 3: How model 1 (i.e., without NMH) differs from model 2 (with NMH) is not clearly shown in the diagram. Please either split the flowcharts or add some description in the caption.

Response: Thanks for your valuable comments. We re-plotted the flowchart to highlight the role of inner model. Additionally, we added description on the difference between basic model and nested model in the caption of Figure 3.

"The basic model (Model I) does not consider NMH from the inner model and utilizes only ten input variables for testing and training, namely: satellite $NO_2$, two temporal variables, and seven meteorological variables. The nested model (Model II) considers the NMH from the inner model as an additional input variable, along with the other ten input variables used for the basic model. Therefore, the nested model utilizes eleven input variables for testing and training: satellite $NO_2$, two temporal variables, seven meteorological variables, and the NMH predictions from the inner model."

**Comment:** Figure 4: Please clarify the meaning of each figure element (dots with colors, lines, etc.).

Response: Thank you very much for your comments. We clarified the meaning of the figure elements in lines 303-306:

> "The red dotted line represents a 1:1 relationship. The solid black line is the line of best fit between the ground-measured $NO_2$ and the satellite-estimated $NO_2$. The scattered dots represent the individual $NO_2$ values for each ground measurement and satellite-based estimation. The color scale ranging from red to blue represents the density of the $NO_2$ values, with red indicating high density and blue representing low density."

**Comment:** Figure 7: Is this figure corresponding to ground-based observations or satellite-based estimates? Is it an average of 8 AM to 3 PM local time or daily average? Please clarify. Also, mark the province if possible so that readers unfamiliar with China can have a better sense of the regions you are referring to.

Response: Thanks very much for your comment. This figure presents satellite-based estimates of the annual average ground-level $NO_2$ concentration, which represents the 24-hour average throughout the year 2021, after bias correction for the missing data issue. We have clarified this information in the caption of the figure (lines 364-368). Additionally, provinces are marked in the figure.

> "Spatial distributions of annual average ground-level $NO_2$ concentrations for 2021 derived from satellite measurements in the study region (left panel) and in the four major urban agglomerations in China (right panel): Beijing-Tianjin-Hebei (BTH), Yangtze River Delta (YRD), Pearl River Delta (PRD), and Sichuan Basin (SCB). This annual average concentration represents the 24-hour average throughout the year of 2021 after the bias correction for the missing data issue."

**Comment:** Figures 9 through 12: What are the vertical bars in each plot? Please clarify.

Response: Thanks for your comment.

(1) The vertical bars in figure 9 (now figure 10), 10 (now figure 11) and 11 (now figure 13) represent one standard deviation. The description is added in the caption of the figures.

(2) The vertical bars in figure 12 (now figure 14) represent whiskers that extend to the most extreme data points within 1.5 times the interquartile range from quartile 1 (25th percentile of data) and quartile 3 (75th percentile of the data). The description is also added in the caption of the figure.

---

## Author Response (AR2)

Dear Editor and Reviewer,

We are grateful for the helpful comments. We have made the modifications in response to comments. Attached is a point-by-point response to the comments. We hope that you and the referees will find the changes satisfactory, and we look forward to hearing from you soon.

**I) Comments from the editor**

I am pleased to inform you that your manuscript is accpeted for publication in ACP after considering the technical corrections given by the referee (see referee report) and the notification to the authors from review file validation (see box above) plus the ones from me listed in the following:

**Comment:** L213-214: "The XGBoost model was implemented in this study to convert columnar measurements into ground-level NO2 concentrations." and L215-216: "In this study, a nested machine learning model was developed to incorporate the NMH to convert columnar 215 measurements into ground-level NO2 concentrations."
Doubling? Please check and rephrase.

**Response:** Thank you for the comment. We have removed the doubling.

**Comment:** Is the XGBoost the machine learning model that you are using? If yes, you should clearly state this and decide if you then write in the following "XGBoost" or "machine learning model". Or are these two different models? Is XGBoost the model that you later (L234-235) denote as model 1 and the nested machine learning model the one you later denote as model 2?

**Response:** Thank you for the comment. Yes, the XGBoost model is the machine learning model that we used in this study. We have trained two models in our study. Model I is a basic XGBoost machine learning model trained without considering NMH. Model II is a nested XGBoost machine learning model, which includes an inner model designed to consider NMH as input parameters. Specifically, within this nested XGBoost machine learning model (Model II), the inner model predicts the NMH, which is then input into the main XGBoost model to predict ground-level $NO_2$ concentrations. We clarify the nested XGBoost machine learning model throughout the manuscript, such as (lines 236-238):

> "To reveal the impacts of the NMH, we compared the performance of the basic XGBoost machine learning model without considering the NMH (Model I) and the nested XGBoost machine learning model after considering the NMH (Model II)."

**Comment:** L349: in further analyses -> in the further analyses

**Response:** Thank you for the comment. We have revised it accordingly (line 356).

**Comment:** L383: Abbreviation AQG has not been introduced? I found it in the introduction, but it would

be good to repeat it here.

**Response:** Thank you for the comment. We have revised and added it in lines 390-393.

"All provinces depicted population exposure levels of $NO_2$ exceeding the Air Quality Guidelines (AQG) levels of 10 μg/m³. Hainan Province had the lowest population-weighted mean $NO_2$ concentrations of 10.57 μg/m³, which closely approached the levels set by the AQG."

**Comment:** Figure 10 caption: "one standard deviation"? Do you mean one sigma standard deviation?

**Response:** Thank you for the comment. It is one sigma standard deviation and we added in the caption (line 410-411).

**Comment:** Figure 11 caption: same here

**Response:** Thank you for the comment. It is one sigma standard deviation and we added in the caption (line 434-435).

**Comment:** L428: average -> averaged

**Response:** Thank you for the comment. We have revised it accordingly (line 437).

**II) Comments from referee:**

The authors have revised the manuscript throughly. I believe it is ready for publication as long as several technical corrections are made.

Minor comments:

**Comment:** Line 23: This nested model was designed to ... and explore its impact on performance. Was this nested model really designed to explore its impact on performance? This statement sounds awkward.

**Response:** Thank you for the comment. We have revised the statement (line 26-27).

"This nested model was designed to directly incorporate NMH into the methodological framework to estimate satellite derived ground-level $NO_2$ concentrations."

**Comment:** Line 27: estimating ground-level NO2 concentration -> ground-level NO2 concentration estimates.

**Response:** Thank you for the comment. We have revised it accordingly (line 30).

**Comment:** Line 27: reducing bias and improving the R2 values to 0.93 ... -> reducing the bias by <a value> and improving the R2 values from <a value> to 0.93 ...

**Response:** Thank you for the comment. We have revised as per suggestion in lines 29-31.

"The inclusion of NMH significantly enhanced the accuracy of ground-level $NO_2$ concentration estimates, i.e., the R² values were improved from 0.73 to 0.93 in 10-fold cross-validation and

from 0.88 to 0.99 in the fully trained model."

**Comment:** Line 136: According to Kim et al. (2020), the nominal spatial resolution of GEMS baseline products is ~7 km x 8 km. However, the GEMS NO2 product has a spatial resolution of ~7 km x 8 km x 2 px (due to spatial bining, i.e., ~ 14 km x 8 km). Please double check if your statement is correct here.

**Response:** Thank you for the comment. We have checked the statement and described it in lines 139-140.

"The nominal spatial resolution of the GEMS $NO_2$ product was 7 km × 7.7 km, by binning two pixels of 3.5 km × 7.7 km each (Ahmad et., 2024)."

**Comment:** Line 261: Suggest replacing semicolons (;) with commas (,) in this sentence.

**Response:** Thank you for the comment. We have replaced the semicolons with commas in the sentence as per suggestion.

**Comment:** Line 263: Is 365 the maximum value or the maximum possible value?

**Response:** Thank you for the comment. 365 is the maximum possible value here. We have revised accordingly in line 267.

**Comment:** Line 280: Looks like you are interpreting m differently here and in Line 263. Try to avoid this as it may confuse the readers. One possible solution is to use one letter for 24 hours of a day, and another letter for 365 days of a year (366 if a leap year).

**Response:** Thank you for the comment. To avoid possible confusion for the readers, we have added a new equation to obtain the annual correction factor and its corresponding description in lines 282-288.

"Similarly, to obtain the annual correction factor, we estimated the ratio between the annual average of all available ground-measured $NO_2$ concentrations for 24 hours and the annual average of ground-measured $NO_2$ when the satellite data was available (Eq. 3).

$$F = \frac{\frac{1}{j}\sum_{i=1}^{j} C_g(i)}{\frac{1}{p}\sum_{i=1}^{p} C_g(i)} \qquad (3)$$

Here, $F$ represents the annual correction factor, $C_g$ represents ground-measured $NO_2$ concentrations, $j$ shows all ground measurements of $NO_2$, and $p$ corresponds to ground measurements of $NO_2$ only when the satellite data was available. For the annual correction factor, the maximum possible value of $j$ index in Eq. 3 is 8760 for one year."

**Comment:** Figure 3: Is 'Tree-n' supposed to be the same as 'Tree-1'? Also, the placement of 'Ground Measured NO2', 'Output', and 'Input' (the bottom one) looks confusing. I suggest placing all inputs at top and te output (ground-level NO2 predictions) at bottom, and reorganizing your arrow/flows accordingly.

**Response:** Thank you for the comment. The trees from Tree-1 to Tree-n could be different. We have

modified the subplot for Tree-n. Further, Fig. 3 has been revised as per suggestion. All the inputs are placed on top and output at bottom.

**Comment:** Figure 13: Vertical bars are overlapping.

**Response:** Thank you for the comment. We have replotted the figure to solve the overlapping issue.

**III) Comments from Notification to the authors:**

Comment 1:

The system indicates that at least one of the (co-)authors is a member of the editorial board of ACP (Fangqun Yu). Please add this information into the *.pdf manuscript under the headline "Competing interests" with the next revision. See more: https://www.atmospheric-chemistry-and-physics.net/submission.html#manuscriptcomposition > 16. Competing interests > 3.

Response: Thank you for the comment. We have added the information accordingly (line 673-674).

Comment 2:

Please ensure that the colour schemes used in your maps and charts allow readers with colour vision deficiencies to correctly interpret your findings (see e.g. F09). Please check your figures using the Coblis Color Blindness Simulator (https://www.color-blindness.com/coblis-color-blindness-simulator/) and revise the colour schemes accordingly.

Response: Thank you for the comment. We have revised the colour schemes of F09, F10, F11 as per suggestions.